# Genetically encoded cell-death indicators (GEDI) to detect an early irreversible commitment to neurodegeneration

Jeremy W. Linsley [1], Kevan Shah [1], Nicholas Castello[1], Michelle Chan[1], Dominik Haddad[2], Zak Doric [2,3], Shijie Wang[1], Wiktoria Leks[1], Jay Mancini[1], Viral Oza[1], Ashkan Javaherian[1], Ken Nakamura [2,4,5], David Kokel[6,7] & Steven Finkbeiner [1,8✉]

Cell death is a critical process that occurs normally in health and disease. However, its study is limited due to available technologies that only detect very late stages in the process or specific death mechanisms. Here, we report the development of a family of fluorescent biosensors called genetically encoded death indicators (GEDIs). GEDIs specifically detect an intracellular $Ca^{2+}$ level that cells achieve early in the cell death process and that marks a stage at which cells are irreversibly committed to die. The time-resolved nature of a GEDI delineates a binary demarcation of cell life and death in real time, reformulating the definition of cell death. We demonstrate that GEDIs acutely and accurately report death of rodent and human neurons in vitro, and show that GEDIs enable an automated imaging platform for single cell detection of neuronal death in vivo in zebrafish larvae. With a quantitative pseudo-ratiometric signal, GEDIs facilitate high-throughput analysis of cell death in time-lapse imaging analysis, providing the necessary resolution and scale to identify early factors leading to cell death in studies of neurodegeneration.

[1] Gladstone Center for Systems and Therapeutics, San Francisco, CA, USA. [2] Gladstone Institute of Neurologic Disease, San Francisco, CA, USA. [3] Neuroscience Graduate Program, University of California, San Francisco, CA, USA. [4] Biomedical Sciences and Neuroscience Graduate Program, University of California, San Francisco, CA, USA. [5] Department of Neurology, University of California, San Francisco, CA, USA. [6] Department of Physiology, University of California, San Francisco, CA, USA. [7] Institute for Neurodegenerative Disease, University of California, San Francisco, CA, USA. [8] Taube/Koret Center for Neurodegenerative Disease, Gladstone Institutes, San Francisco, CA, USA. ✉email: sfinkbeiner@gladstone.ucsf.edu

Neurodegenerative diseases such as Parkinson's disease (PD)[1,2], Huntington's disease (HD)[3–7], frontotemporal dementia (FTD), Alzheimer's disease (AD), and amyotrophic lateral sclerosis (ALS)[8,9] are characterized by progressive neuronal dysfunction and death, leading to a deterioration of cognitive, behavioral or motor functions. In some cases, neuronal death itself is a better correlate of clinical symptoms than other pathological hallmarks of disease such as Lewy bodies in PD[10], or β-Amyloid in AD[11], and can be used to effectively characterize the relationship of an associated disease phenotype with degenerative pathology[12,13]. Using neuronal death as a consistent and unequivocal endpoint, longitudinal single-cell analysis can be performed on model systems to reveal the antecedents and forestallments of cell death[14,15]. Together with statistical tools used in clinical trials that account for variability, stochasticity, and asynchronicity amongst individuals within a cohort, it is possible to regress premortem phenotypic markers of neurodegeneration[16], determine which are beneficial, pathological, or incidental to degeneration, and quantify the magnitude of their contribution to fate. For instance, although inclusion bodies are a hallmark of disease in HD and Tar DNA binding protein 43 (TDP43) ALS, their presence appears to be more consistent with a coping mechanism rather than a causative factor, suggesting clinical intervention to inhibit inclusion body formation could be a misguided approach[3,8]. While there is an ongoing debate about the relative contribution of neuronal dysfunction prior to neuronal death to the clinical deficits that patients exhibit, it is clear that neuronal death marks an irreversible step in neurodegenerative disease. Thus, neuronal death is an important, disease-relevant phenotypic endpoint that is important to understanding neurodegeneration, characterizing the mechanisms of neurodegenerative disease, and developing novel therapeutics.

Nevertheless, precisely determining whether a particular neuron is alive, dead, or dying can be challenging, particularly in live-imaging studies. Vital dyes, stains, and indicators have been developed to selectively label live or dead cells and neurons in culture[17], but the onset of these signals may be delayed until long after a neuron has shown obvious signs of degeneration[12]. Additionally, long-term exposure to exogenous dyes can increase the risk of accumulated exposure toxicity, negating their ability to noninvasively provide information on cell death. Many assays can distinguish between cell death pathways such as apoptosis or necrosis[13] and are conducive to longitudinal imaging[18,19], but these typically require a priori knowledge of which cell death pathway is relevant, limiting their utility in neurodegenerative disease, which often involves a spectrum of neuronal death mechanisms[20]. Moreover, there is often high interconnectivity of signaling molecules across cell death pathways, and cells that begin to die by one cell death pathway may resort to a different one if the original pathway is blocked[21], which can confound analyses. Furthermore, the interpretation and accuracy of apoptotic markers can vary based on the specific cellular system, and some are associated with reversible processes, meaning multiple assays must be used in parallel to unambiguously characterize the precise extent of death within a sample[13,21]. For these reasons, an early and sensitive cell-death pathway agnostic marker is often necessary to give a complete and unbiased readout of cell death. In live-imaging experiments, the loss of fluorescence of transfected neurons, indicating the rupture of the plasma membrane, has been shown to clearly mark neuronal death[1,3], but fluorescent debris often persists for days after initial morphological signs of death and decay occur, limiting the ability to identify the precise time of death or introducing human error in the scoring of neuronal death by morphology[12]. In summary, without a strict criterion of what constitutes a "point of no return" at which a neuron's fate is unambiguously sealed,

investigation of the causative factors that precede cell death remains challenging.

Although dyes have been used to detect neuronal death in vivo[22–24], the permeability of dyes throughout tissue is inconsistent, making quantification difficult. Genetically encoded fluorescent proteins have greatly facilitated the ability to track single neurons within culture[3] and in tissue[25] over time. Furthermore, genetic targeting allows labeling of specific cell subtypes, as well as simultaneous expression of other biosensors, perturbagens, or activators[15]. Some of the most commonly used biosensors in neuroscience are the genetically-encoded $Ca^{2+}$ indicators (GECIs), including the yellow cameleons and GCaMPs/pericams[26,27]. Based on the fusion of circularly permuted fluorescent proteins such as GFP with the $Ca^{2+}$ binding M13-calmodulin domain, GECIs are relatively bright biosensors with low toxicity in neurons[28]. GECIs are used to detect either relative or absolute $Ca^{2+}$ levels or neuronal circuit activity within neurons in culture[29], in tissue[30], within immobilized animals in virtual environments[31,32], and even within freely moving animals[33,34]. The category of GECIs has been diversified and further optimized through the use of alternate fusion proteins to the M13-calmodulin domain as well as targeted mutagenesis[35,36]. Recently, Suzuki et al. engineered endoplasmic reticulum (ER)-targeted calcium-measuring organelle-entrapped protein indicators (CEPIAer) variants that emit either green, red, or blue/green fluorescence to specifically detect $Ca^{2+}$ release events from the ER[36]. In contrast to previously developed ER $Ca^{2+}$ indicator dyes, CEPIAs and recently engineered derivatives[37] can be genetically and subcellularly targeted and are capable of long-term imaging over the lifetime of the neuron, enabling measurement of the full range of calcium dynamics within single neurons over time.

Here, we introduce a class of GECIs for the detection of cell death in neurons that we call genetically encoded death indicators (GEDIs). We show that GEDIs can robustly indicate the moment when a neuron's ability to maintain $Ca^{2+}$ homeostasis is lost and cannot be restored, providing an earlier and more acute demarcation of the moment of death in a degenerating neuron than previously possible. In combination with a second fluorescent protein fused with a self-cleaving P2a peptide, pseudo-ratiometric GEDIs are easily quantifiable in high-throughput, give a highly reproducible signal, and are amenable to long-term imaging. GEDIs can also be targeted to specific neuronal subtypes for imaging in vivo. These data establish GEDIs as important tools for studying the time course of neurodegeneration, providing previously unobtainable delimitation and clarity to the time course of cell death.

## Results

**Development of genetically encoded death indicators**. GECIs such as GCaMP6f have been engineered to increase fluorescence in response to fluctuations in the range of cytosolic $Ca^{2+}$ concentrations that occur during neuronal firing (Fig. 1A). The CEPIA GECIs have been engineered with elevated $K_d$ to detect $Ca^{2+}$ transients in organelles such as the ER or mitochondria that contain higher $Ca^{2+}$ (Fig. 1A)[36]. We reasoned that removing the ER retention signals and allowing CEPIA variants to localize to the cytosol would render a GECI that was not responsive to activity-based $Ca^{2+}$ transients, but that would increase in fluorescence intensity when cytosolic $Ca^{2+}$ levels approached those of intracellular organelles or the extracellular milieu, which would constitute a catastrophic event for the neuron. We named these reengineered indicators GEDIs. In rat cortical primary neurons, brief electrical field stimulation (3 s at 30 Hz) increased fluorescence within cells expressing GCaMP6f, but not in those expressing the red GEDI variant (RGEDI) (Fig. 1B–D). The

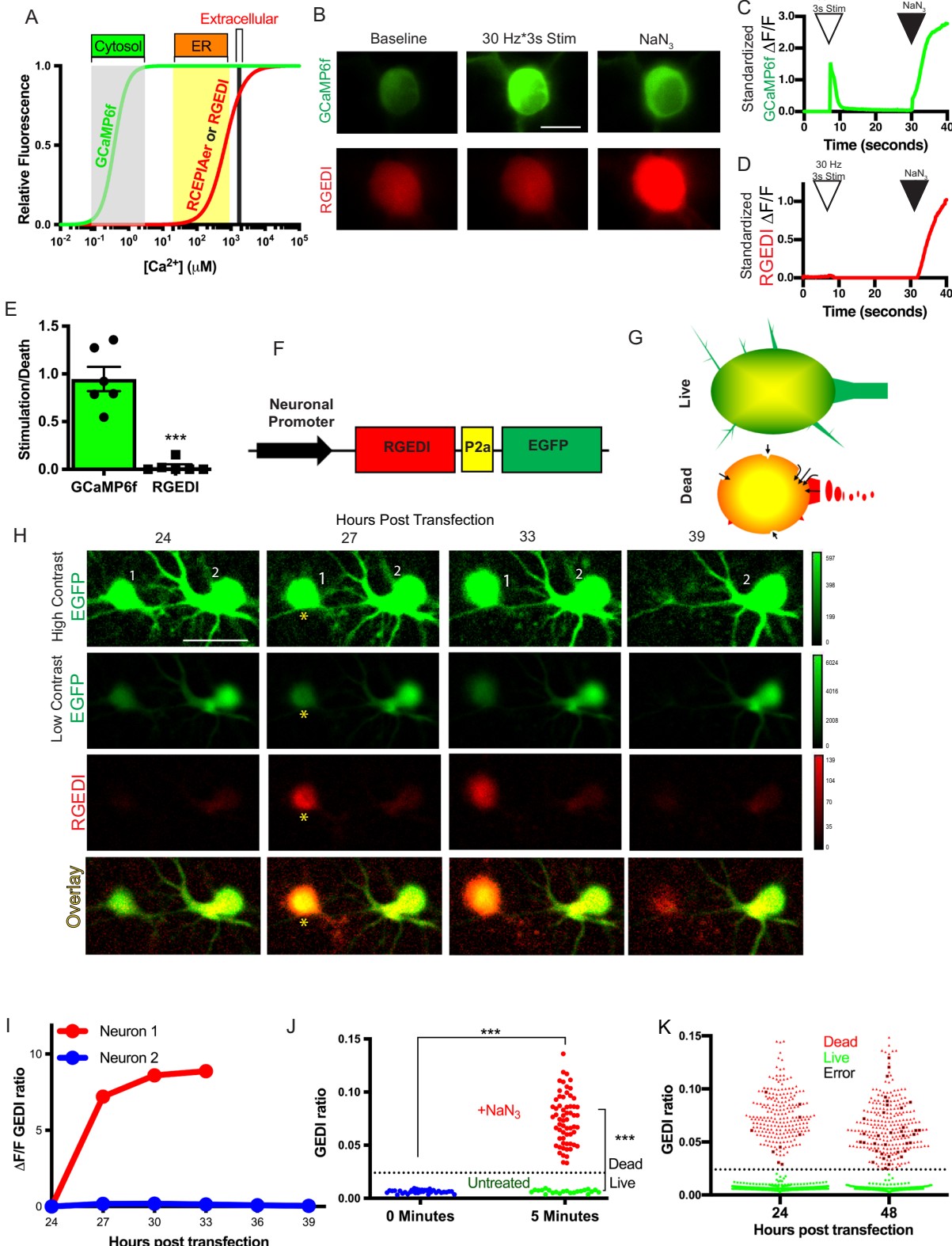

addition of NaN$_3$, to a known cytotoxin which induces neuronal death, caused increased fluorescence in cells expressing either GCaMP6f or RGEDI (Fig. 1B–D). The peak GCaMP6f fluorescence response after stimulation was nearly identical to the fluorescence response to NaN$_3$ treatment; in contrast, the ratio of stimulation to death response in RGEDI expressing cells was close to 0, indicating that the RGEDI sensor preferentially responds to

death (Fig. 1E). Removal of extracellular Ca$^{2+}$ abrogated the fluorescence response to NaN$_3$ treatment, indicating the primary increase in cytosolic Ca$^{2+}$ required influx from the extracellular space (Supplementary Fig. 1). To further optimize the RGEDI construct, we appended sequence encoding a porcine teschovirus-1 2a (P2a) "self-cleaving peptide" and EGFP, allowing normalization of the RGEDI signal to EGFP expression (GEDI ratio).

**Fig. 1 GEDI detects the death of neurons. A** Relative fluorescence of GECI's GCaMP6f[79] and RCEPIA[36]/RGEDI across $Ca^{2+}$ concentrations present in the cytosol, endoplasmic reticulum, and extracellular milieu modeled from previously reported Hill coefficients and $K_d$ values. **B** Representative fluorescence images of rat primary cortical neurons transfected with GCaMP6f or RGEDI at baseline, during 30 Hz × 3 s field stimulation, or after 2% $NaN_3$ treatment, scale = 10 μm. The experiment was repeated six times for each condition with similar results. **C, D** Representative trace of the time course of standardized ΔF/F fluorescence after 30 Hz × 3 s stimulation (open arrowhead), and after $NaN_3$ treatment (black arrowhead) of GCaMP6f (**C**) and RGEDI (**D**) expressing neurons. **E** To determine whether RGEDI signals were specific for cell death and not confounded by physiological $Ca^{2+}$ transients, the ratio of the maximum signal from electrical stimulation to toxin treatment per neuron are shown, demonstrating that GEDI does not respond to physiological $Ca^{2+}$ transients ($n = 6$ cells/group, 1 cell per coverslip, *** Unpaired, two-tailed $T$-test, $p < 0.0001$). Error bars represent SEM. **F** Design of RGEDI-P2a-EGFP cassette for pseudo-ratiometric expression in neurons. **G** Illustration of color change in red:green image overlay expected in a live versus dead neuron. Live neurons have EGFP (green) and basal RGEDI (red) fluorescence (overlaid as yellow) within the soma of the neuron, surrounded by green fluorescence that expands through the neurites. Dead neurons display yellow fluorescence throughout, with edges of red fluorescence around the soma and throughout degenerating neurites as extracellular $Ca^{2+}$ permeates the membrane (arrows). **H** Time course images of rat primary cortical neurons expressing RGEDI-P2a-EGFP at 24–39 h post transfection (hpt). Neuron 1 shows characteristic morphology features of life through 27 hpt at which time it also shows elevated GEDI ratio (yellow asterisk), followed by loss of fluorescence at 39 hpt. Neuron 2 remains alive through time course, scale = 20 μm. Color scales are annotated in arbitrary units for each color channel. **I** Quantification of change in GEDI ratio of Neurons 1 and 2 in (**H**). **J** Quantification of GEDI ratio in rat primary cortical neurons before (blue dots) and from a separate well 5 min after $NaN_3$-induced neuronal death (green dots = live, red dots = dead). The Black dotted line represents the calculated GEDI threshold. (*** One-tailed $T$-test $p < 0.0001$). **K** Quantification and classification of death in neuronal cultures at 24 and 48 h after co-transfection of RGEDI-P2a-EGFP and an N-terminal exon 1 fragment of Huntingtin, the protein that causes Huntington's disease, with a disease-associated expansion of the polyglutamine stretch (HttEx1Q97) to induce neuronal death using the derived GEDI threshold from (**J**) to define death. Independently, neurons were scored as dead (red), or live (green) by eye using EGFP at 24, 48, and 72 post transfection. Neurons that were classified as live by eye but above the GEDI threshold and classified as dead at the subsequent time point were called human errors (black).

This facilitated simple detection of the moment of neuronal death based on a cell's color change when the green and red channels are overlaid (Fig. 1F, G).

To assess the ability of RGEDI to detect death in live-cell imaging, we used automated longitudinal microscopy[3] to image individual neurons transfected with hSyn1:RGEDI-P2a-EGFP repeatedly at 3 h intervals (Fig. 1H, I). Neurons that died over the course of imaging were marked by clear fragmentation of morphology in the EGFP channel, followed by the disappearance of the debris[3,38,39]. Once a cell died, the increase in GEDI ratio remained stable over the course of imaging until the disappearance of the debris (Fig. 1H, I). In some cases, an increase in the GEDI ratio preceded obvious morphology changes (Fig. 1H). GEDI signal correlated with and often preceded standard markers for neuronal death such as TUNEL[40], ethidium homodimer D1 (EthD1)[41], propidium iodide (PI)[42], or the human curation of the morphology[3] (Supplementary Fig. 2). GEDI signal also preceded markers of apoptotic death such as Caspase3/7 signal (Supplementary Fig. 3). Due to the large separation of the GEDI ratio between live and dead neurons, we established a formal threshold of death for the GEDI ratio that could be used to quantify the amount of cell death in high throughput. Rat primary cortical neurons were transfected with hSyn1:RGEDI-P2a-EGFP in a 96-well plate and the GEDI ratio was derived from each well before and after a subset was exposed $NaN_3$ (Fig. 1J). After 5 min, all neurons exposed to $NaN_3$ showed an increased GEDI ratio compared to neurons before treatment and those not exposed to $NaN_3$ (Fig. 1J). From these data, a GEDI ratio corresponding to the threshold of death (GEDI threshold) was calculated according to Eq. 1 (see Methods).

Automated microscopy was then performed at 24 h intervals for four days on the remaining 94 wells of the plate, and the previously derived GEDI threshold was used to assess the spontaneous neuronal death of hSyn1:RGEDI-P2a-EGFP-transfected neurons (Fig. 1K). In parallel, neuronal death was assessed by manual curation based on the abrupt loss of neuronal fluorescence over time. All neurons identified as live by manual curation had a GEDI ratio below the GEDI threshold, and most neurons identified as dead by manual curation had a GEDI ratio above the GEDI threshold (Fig. 1K). The few neurons classified as live by manual curation that contained a GEDI ratio above the threshold were recognized in hindsight to be difficult to classify

based on morphology alone. Furthermore, each of these neurons could later be unequivocally classified as dead due to loss or fragmented pattern of fluorescence at the next imaged time point (Supplementary Fig. 4A′, A″), suggesting an error or inability of humans to correctly classify, and demonstrating GEDI is a more acute and accurate means of classifying neuronal death than manual curation. To examine the possibility that a temporarily increased GEDI ratio signal may give a false positive death indication, a large longitudinal data set of time-lapse GEDI images containing 94,106 tracked neurons across multiple imaging conditions was generated (Supplementary Table 1). Across all 32 longitudinal experiments, a consistent GEDI threshold indicating cell death could be established (Supplementary Fig. 4B, C). Only 0.28% of all neurons exhibited a GEDI ratio above the threshold at a one-time point that subsequently decreased below the death threshold (Supplementary Fig. 4D). Upon closer examination of those 304 neurons, each was subsequently found to have an automated segmentation artifact, which distorted the ratio of RGEDI to GFP or a tracking artifact that confused objects, rather than decreased RGEDI signal in relation to GFP (Supplementary Fig. 4E′, E″). No neurons were found to "die twice," as would be indicated by two fluctuations in the GEDI ratio above the death threshold. These data show that the GEDI signal is unlikely to increase above the GEDI threshold in a neuron that is not dead. Maintaining $Ca^{2+}$ homeostasis is thought to be important for all cell types, and GEDI signal also differentiated live and dead cells when expressed in HEK293 cells (Supplementary Fig. 5). Therefore, we conclude that the GEDI biosensor specifically signals an early and irreversible commitment to degeneration and death and can serve as ground truth for quantifying cell death.

**Automated identification of toxin-resistant and sensitive subpopulations of neurons.** We predicted that by combining GEDI and time-lapse imaging, we would be able to monitor a heterogeneous death process and identify time-resolved subpopulations of neurons with differing sensitivities. Glutamate is the most common neurotransmitter in the brain[43], but glutamate excess occurs in neurodegenerative disease and has been shown to be toxic to specific subpopulations of neurons[44]. Glutamate toxicity induces either apoptosis[44] or necrosis[45], and a reliable death sensor capable of detecting either death type can facilitate an

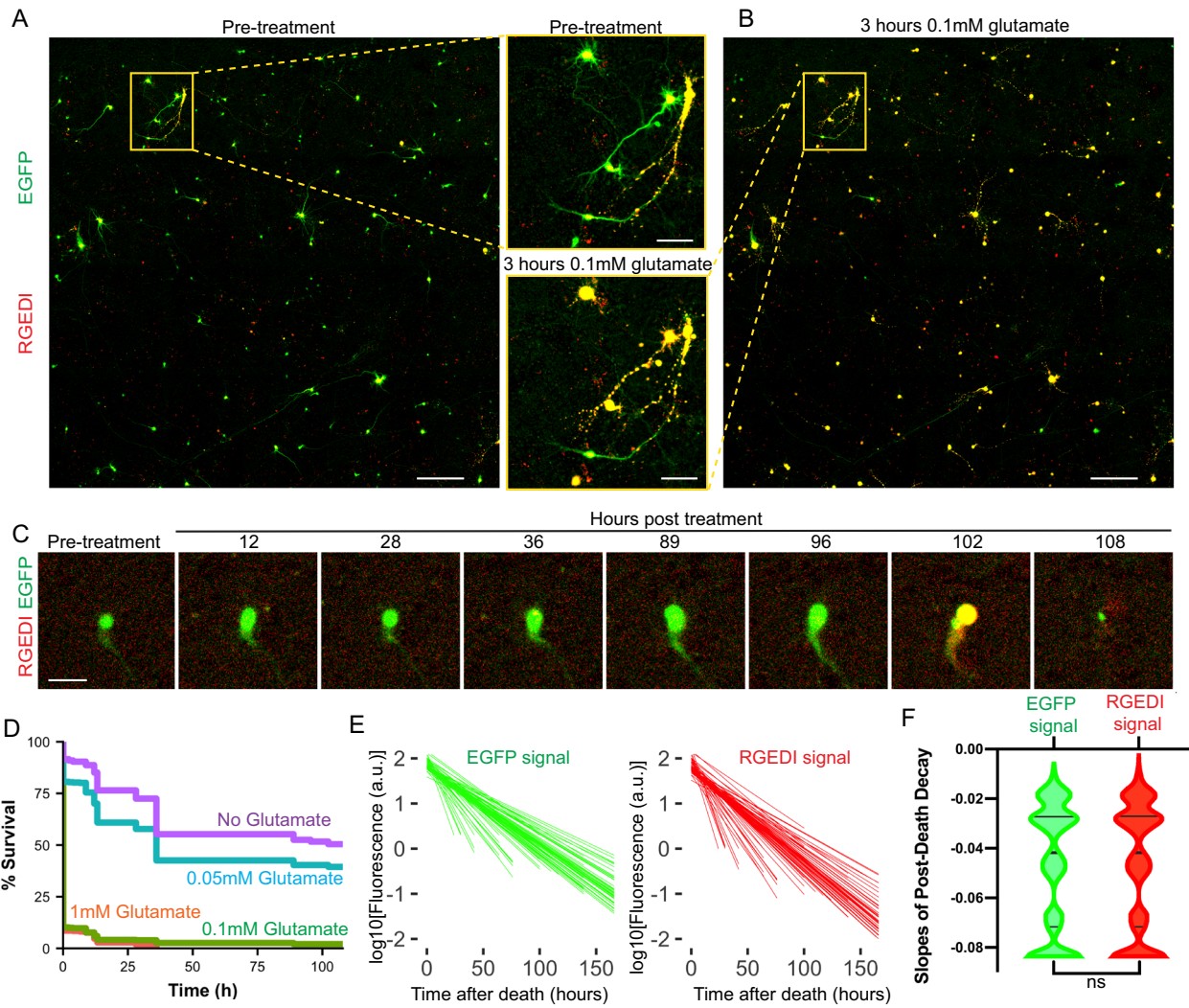

**Fig. 2 Detection and characterization of subpopulations of neurons resistant or sensitive to glutamate treatment using GEDI. A** Representative two-color overlay image from a well of rat primary cortical neurons expressing RGEDI-P2a-EGFP before treatment with 0.1 mM glutamate (Scale = 400 μm) and zoom-in of two individual neurons within the yellow box (Scale = 100 μm). Live neurons appear green/yellow, dead neurons appear yellow/red. **B** Representative two-color overlay image from neurons 3 h after treatment with 1 mM glutamate and enlargement of the same two neurons within the yellow box in (**A**). Scale = 100 μm. **C** Time-course images of a neuron before and after exposure to 0.1 mM glutamate that survives until 96 h post-treatment (Scale = 50 μm). Experiments **A–C** were repeated on 16 wells with similar results. **D** Kaplan–Meier plot of neuron survival after exposure to 1, 0.1, 0.01 mM, and 0 glutamate ($n = 699, 585, 527, 476$). **E** Linear regressions of decay of EGFP (left) and RGEDI (right) after neuronal death marked by GEDI signal above the threshold. **F** Slopes of decay of EGFP and RGEDI signals are not different (Mann–Whitney, two-tailed, ns not significant, $p = 0.93$, $n = 5513$). Horizontal lines in the violin plots represent quartiles and median.

unbiased accounting of toxicity. In principle, RGEDI should be able to detect all cell death events, as it detects loss of $Ca^{2+}$ homeostasis rather than a specific substrate of a cell death pathway. To assess glutamate toxicity, rat primary neurons were transfected with hSyn1:RGEDI-P2a-EGFP and followed with automated microscopy after exposure to different levels of glutamate (Fig. 2A–C). A GEDI threshold was used to define dead neurons, and Kaplan–Meier survival curves were generated for the time course of imaging (Fig. 2D). While over 90% of neurons died within 3 h of exposure to 0.1 mM or 1 mM glutamate, recognizable by the stark change in composite GEDI color of images of wells (Fig. 2A, B), sparse neurons resistant to those treatments could be identified by their low GEDI ratio that remained alive in the culture long after the initial wave of death. Some neurons remained alive to the end of the 108 h imaging window in the presence of glutamate (Fig. 2C, D).

It is known that GFP and RFP are differentially sensitive to lysosomal hydrolases due to the difference in pH stability of GFP and RFP, and that differential sensitivity has been exploited to develop tandem tag biosensors to measure autophagy[46,47]. Since $Ca^{2+}$ is known to activate certain proteases[48], we wondered if differential degradation rates of RGEDI and EGFP fluorescence signals in dead neurons could cause the GEDI ratio to fluctuate and under- or overrepresent death. To investigate, we characterized the decay rate of EGFP and RGEDI signals after rapid death from glutamate toxicity. In dead neurons, the relative fluorescence of each protein decayed at equivalent rates ($t_{1/2} = 20.45$ h RGEDI, 20.73 h EGFP), indicating the activated GEDI ratio is a stable indicator of death across long time intervals of imaging during which neuronal debris remains present (Fig. 2E, F). These data suggest that a GEDI is a powerful tool to accurately identify live neurons within a culture in which extensive death has occurred.

**Automated survival analysis of multiple neurodegenerative disease models in different species with GEDI.** Neurodegenerative disease-related neuronal death is associated with a spectrum of death mechanisms including apoptosis, necrosis, excitotoxicity, and autophagic cell death[20,49], necessitating the use of a death indicator of all types of cell death for effective and unbiased detection of total death. To test the effectiveness of GEDI across neurodegenerative disease models, hSyn1:RGEDI-P2a-EGFP was cotransfected into rat cortical primary neurons with pGW1:HttEx1-Q25 or pGW1:HttEx1-Q97, pCAGGs:α-synuclein, or pGW1:TDP43 to generate previously characterized overexpression models of HD[3], PD[39], and ALS or FTD[50], respectively (Fig. 3A). Each model has been associated with multiple types of cell death to varying degrees[40,51–53]. In each model, neurons with characteristic yellow overlays of the RGEDI and EGFP channel could be detected at 24 h (Fig. 3A–A″). GEDI ratios for neurons in each model and control at each time point were quantified, and a GEDI threshold was calculated using a subset of neurons designated dead or live by manual curation (Fig. 3A, B). GEDI ratios from dead neurons in disease models were lower than GEDI ratios from their respective controls, likely due to the combined effects of reduced total exogenous protein expression observed in each disease model compared to control, and lower separation of GEDI ratio between live and dead neurons at lower RGEDI-P2a-EGFP expression levels (Supplementary Fig. 6). High expression of RGEDI-P2a-EGFP likely correlates with high expression of co-transfected disease-causing protein, leading neurons with high exogenous protein expression levels to die and disappear sooner, causing underrepresentation of high expression of RGEDI-P2a-EGFP in disease models (Supplementary Fig. 6). Nevertheless, a clear separation of populations of live and dead neurons could still be observed in each case (Fig. 3B–B″). Using the labels generated from the GEDI threshold a cumulative risk-of-death (CRD), a statistical measure of survival used in clinical studies[54], was generated showing significant toxicity of each model compared to controls as previously reported (Fig. 3C). This showed that GEDI can be used to report toxicity over time across a variety of neurodegenerative disease models.

Cell-based overexpression models of neurodegeneration can be difficult to interpret because protein expression above physiological levels can introduce artifacts, which could also affect GEDI quantification. Neurons derived from induced pluripotent stem cells (iPSC) have the advantage of maintaining the genomic variants of the patients from whom the cells came, facilitating the modeling of neurodegenerative diseases[55]. To test whether GEDI can detect death in an iPSC-derived model of the neurodegeneration in which the endogenous disease-causing protein is expressed at physiologically relevant levels, motor neurons (MNs) were derived from iPSCs from normal control patients and compared to neurons derived from patients with a D90A SOD1 mutation, which has been shown to cause ALS[56]. iPSC MNs generated from patients' fibroblasts that carry the D90A SOD1 mutation have been previously shown to model key pathologies associated with ALS, such as neurofilament-containing inclusions and axonal degeneration, though a clear survival phenotype using a CRD to evaluate toxicity over time has not been established[57]. Neurons were transfected with hSyn1:R-GEDI-P2a-EGFP after 19 days of differentiation and imaged every 12 h with automated microscopy (Fig. 3D–F). The GEDI ratio was quantified and a GEDI threshold was derived and used to generate a CRD plot, which showed that SOD1-D90A–containing neurons exhibited an increased risk of death compared to controls, with a CRD of 1.26 (Fig. 3G). These data show that GEDI can be used to automatically detect neuronal death and derive CRDs from human neurons.

**Development of an expanded family of GEDI sensors.** To expand the applications of the GEDI biosensor, we tested other GEDI variants with alternative characteristics that could be useful in different experimental contexts. For example, reliance on green and red emission spectra for death detection with RGEDI-P2a-EGFP restricts the ability to concurrently image other biosensors whose spectra overlap, limiting the opportunity to investigate other co-variates of disease[14]. Accordingly, we engineered RGEDI-P2a-3xBFP, so that death can be reported during simultaneous imaging of green biosensors. RGEDI-P2a-3xBFP showed a significant increase in signal after death compared to RFP-P2a-EGFP and the rate of signal increase following exposure to NaN$_3$ was not different between RGEDI-P2a-3xBFP and RGEDI-P2a-EGFP or GCaMP6f-P2a-mRuby[58] (GEDI ratio = GCaMP6f/mRuby) (Fig. 4A–D ANOVA $p = 0.44$).

Next, we tested a recently engineered ER Ca$^{2+}$ sensor based on the GCaMP GECI called GCaMP6-150, which was recently reported to have an excellent dynamic range[37] and higher affinity for Ca$^{2+}$ than RGEDI (Fig. 4E). We generated a GEDI sensor based on the GCaMP6-150 template by removing the ER signaling peptides from the GCaMP6-150 cassette and combining it with a P2a peptide and mApple to generate GC150-P2a-mApple. As expected, cells expressing GC150-P2a-mApple showed a significantly increased GEDI ratio (GEDI ratio = GC150/mApple) after NaN$_3$-induced death (Fig. 4D–F). Similar to RGEDI-P2a-EGFP, GC150-P2a-mApple did not increase in signal in response to field stimulation of 30 Hz (Supplementary Fig. 7), indicating GC150 signal only increases in response to irreversible high levels of Ca$^{2+}$ but not normal Ca$^{2+}$ transients that occur within neurons.

GECIs targeted to the nucleus have been shown to increase the resolution and duration of signal in whole-brain studies of zebrafish[59]. Therefore, we generated GEDI constructs with nuclear localization signals (NLS): GC150-NLS-P2a-mApple-NLS and RGEDI-NLS-P2a-EGFP-NLS (Fig. 4F, G). Each nuclear-localized GEDI showed a similar significant increase in GEDI ratio upon NaN3 treatment, corresponding to the death of neurons (Fig. 4D, G, H). The kinetics of the responses were not different between any of the GEDI variants (ANOVA $p = 0.65$), indicating a similar ability to detect death across imaging situations in which different versions of GEDI are needed (Fig. 4D). These data demonstrate that the GEDI approach offers an acute, versatile, and quantitative method to detect neuron death in time-lapse imaging.

**GCaMP acutely reports death in vivo.** Many zebrafish larvae models of neurodegeneration have been developed, in part to take advantage of their unique characteristics, including translucent skin and the ability to be immobilized for long periods of time, that make them amenable to live imaging[60,61]. However, it has not been possible to acutely detect neuronal death and characterize the preceding events in vivo with time-lapse imaging in these models, limiting the characterization of neuronal death to static snapshots of single time points[62–64].

We sought to develop a platform to visualize neuronal death longitudinally in vivo with GEDI by adapting our automated four-dimensional (4D) longitudinal single-cell tracking microscopy platform[25] to in toto longitudinal imaging of live zebrafish larvae over multiple days (Supplementary Fig. 8). Larvae at 72 h postfertilization were anesthetized in tricaine, immobilized in low melting point agarose in 96 well optical ZFplates (Diagnocine), where they can typically remain alive for 120 h as assayed by heartbeat. Automated confocal microscopy was used to repeatedly image each fish in three dimensions at specified intervals, generating 4D images of each fish in an array (Supplementary

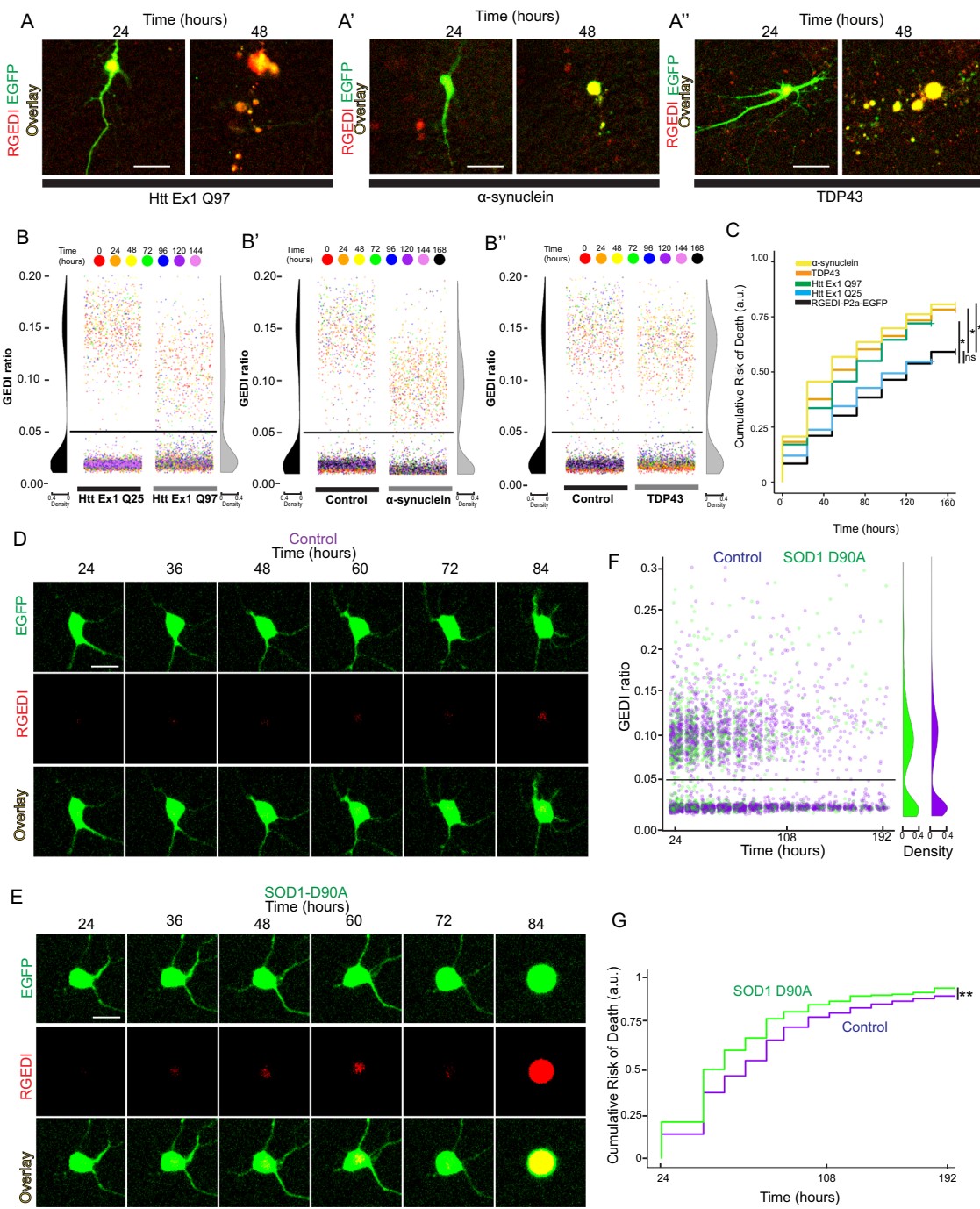

**Fig. 3 Detection of death in neurodegenerative disease models with GEDI. A** Representative two-color overlay images of rat primary cortical neuron at 24 and 48 h after transfection co-expressing RGEDI-P2a-EGFP and HttEx1Q97 (A'), α-synuclein (A''), or TDP43 (A'''). The GEDI ratio identifies each neuron as live at 24, but dead at 48 h post transfection. Scale = 25 µm. **B** Quantification of GEDI ratio during longitudinal imaging across 168 h of live culture of neurons expressing HttEx1Q97 (B), α-synuclein (B'), or TDP43 (B'') with GEDI thresholds at 0.05 for each data set. Dots are color-coded for time post imaging. **C** Cumulative risk-of-death of HttEx1Q97 (HR = 1.83, 95% CI = 1.67–2.01, Cox proportional hazard (CPH) *p < 0.001), HttEx1Q25 (HR = 1.07, 95% CI = 0.99–1.15, ns not significant, p = 0.08), α-synuclein (HR = 1.73, 95% CI = 1.58–1.89, CPH *p < 0.001), TDP43 (HR = 1.77, 95% CI = 1.6–1.94, CPH *p < 0.001), and RGEDI-P2a-EGFP alone (control) generated from GEDI ratio quantification and classification against the GEDI threshold. The number of neurons in Control = 1670, HttEx1Q25-CFP = 1333, HttEx1Q97-CFP = 668, TDP43 = 610, and α-synuclein = 743. **D** Representative time-lapse imaging of a control iPSC motor neuron expressing RGEDI-P2a-EGFP that survives throughout imaging. Scale = 25 µm. **E** Representative time-lapse imaging of a SOD1 D90A iPSC motor neuron that is dead by GEDI signal at 84 h of after transfection. Scale = 25 µm. **F** Quantification of the GEDI ratio of Control and SOD1 D90A neurons and the derived GEDI threshold at 0.05. **G** CRD plot of SOD1-D90A and Control (95% CI = 1.11–1.44, CPH p < 0.0001, number of neurons in Control = 714, and SOD1 D90A = 363).

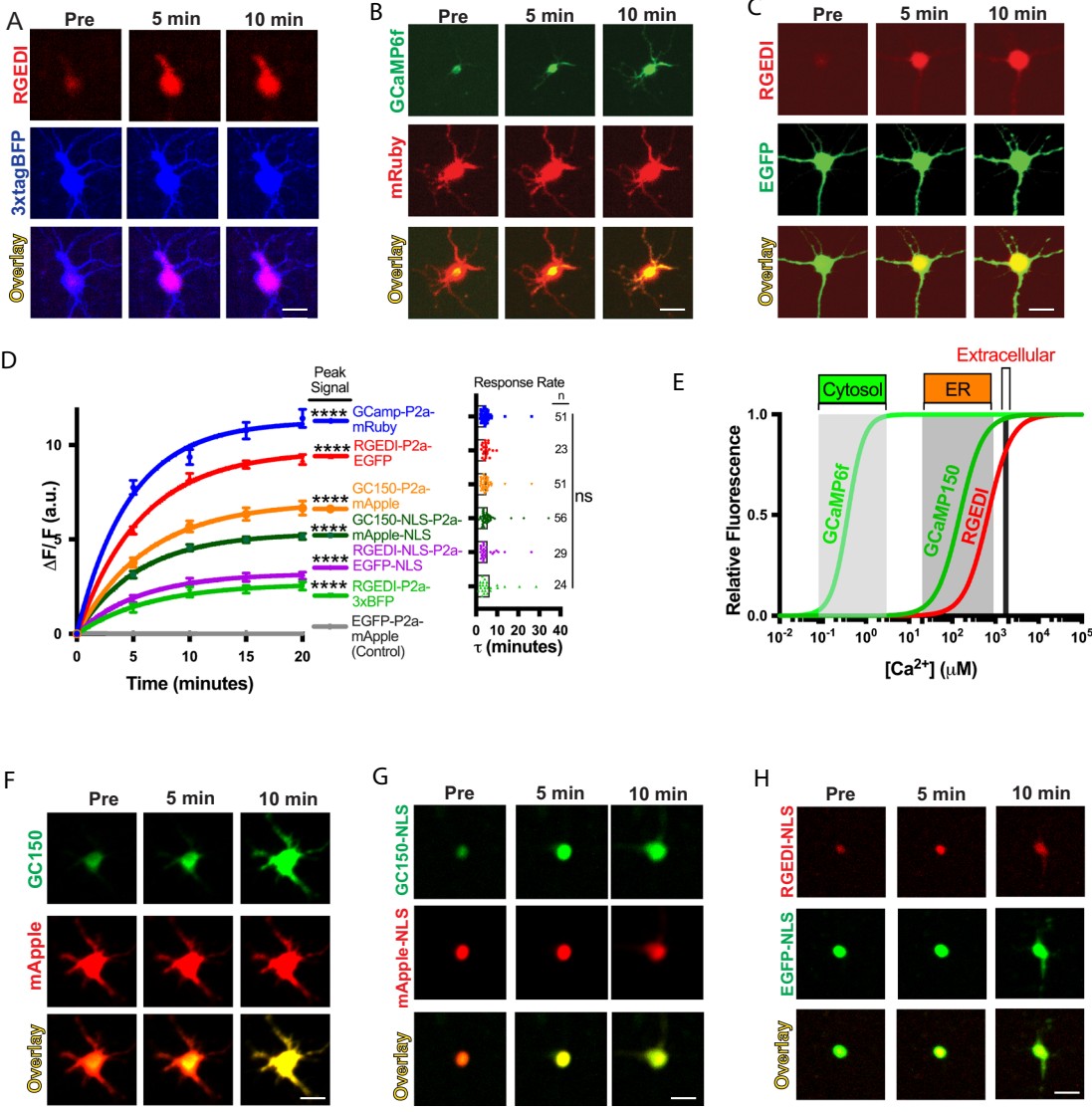

**Fig. 4 Comparison of GEDI variants to detect neuronal death. A–C** Representative images of rat primary cortical neurons expressing RGEDI-P2a-3xBFP (**A**), mRuby-P2a-GCaMP6f (**B**), or RGEDI-P2a-EGFP (**C**), before, 5 min and 10 min after exposure to NaN$_3$. **D** Quantification of the peak signal and response rate of signal increase ($\tau$) from fitted nonlinear regressions of increases in fluorescence signals over time from variants of the GEDI biosensor: RGEDI-P2a-3xBFP ($n = 23$), mRuby-P2a-GCaMP6f ($n = 52$), RGEDI-P2a-EGFP ($n = 41$), GC150-P2a-mApple, RGEDI-NLS-P2a-EGFP-NLS, and GC150-NLS-P2a-mApple-NLS ($n = 40$), compared to EGFP-P2a-mApple ($n = 18$). Error bars represent SE, ANOVA Tukey's ****$p < 0.0001$; ns not significant, $n$ values represent individual neurons sampled across at least three independent wells. **E** Relative fluorescence of GECI's GCaMP6f[79] and RCEPIA[36]/RGEDI and GCaMP150ER[37]/GC150 across Ca$^{2+}$ concentrations modeled from previously reported Hill coefficients and K$_d$ values. **F** Representative images of rat primary cortical neurons expressing GC150-P2a-mApple, **G** GC150-NLS-P2a-mApple-NLS, and **H** RGEDI-NLS-P2a-EGFP-NLS before, 5 min, and 10 min after exposure to NaN$_3$. Error bars represent SEM. Scale = 25 µm. Experiments in **A**–**C** and **F**–**H** were repeated at least 18 times with similar results.

Fig. 8). To evoke neuronal death, the inducible cell ablation protein nitroreductase (NTR)[65] was expressed in MNs using the *mnx1* promoter[66], and 10 µM of metronidazole (MTZ), a harmless prodrug that is activated by NTR, was added to the zebrafish media. By 24 h after the addition of MTZ, some MN axons became clumped and MN cell bodies looked fragmented compared to DMSO, yet no difference in MN axon area was detected (Supplementary Fig. 9A–C). By 48 h after the addition of MTZ, MN axons appeared to retract, and a reduction in axon area could be detected, yet somas and/or debris from the MNs remained in the spinal cord (Supplementary Fig. 9A, B). In contrast, non-immobilized mnx:Gal4; UAS:NTR-mCherry;UAS:EGFP zebrafish larvae became immotile upon incubation with MTZ for 24 h, swimming no more than larvae in which the neuromuscular junction has been blocked with botulinum toxin

(UAS:BoTx-EGFP)[67] (Supplementary Fig. 9D–F), indicating that 24 h incubation in MTZ is sufficient to functionally ablate MNs. Neuronal death at 24 h post MTZ was also confirmed with the use of PhiPhiLux G1D2a live fluorescent reporter of caspase-3-like activity[23,68] (Supplementary Fig. 10A–C). These data suggest that an acute marker for neuronal death is required to monitor neurodegeneration in vivo that more accurately distinguishes live neurons from functionally ablated neurons.

GCaMP is commonly and widely used in zebrafish for studies of neuronal activity and functionality[69], and transgenic lines with GCaMP expression in the nervous system are widely available, making it easy to apply to studies of neuronal death. GCaMP signal due to endogenous Ca$^{2+}$ transients in MNs is not distinguishable from signal due to loss of membrane integrity associated with neuronal death (see Fig. 1). However, we found

that GCaMP can be made to function as a GEDI within the zebrafish by blocking endogenous neuronal activity with the use of tricaine, an anesthetic that blocks voltage-gated channels in the nervous system[70] even when the motor swimming circuit was activated by application of 0.1% acetic acid (AcOH), which stimulates swimming[71] (Supplementary Fig. 11A–C). In contrast, with the use of a muscle contraction blocker 4-methyl-*N*-(phenylmethyl)benzenesulfonamide (BTS), which immobilizes larvae but does not block neuronal activity[72], GCaMP7 calcium transients are still present (Supplementary Fig. 11C–E). GCaMP7 signal significantly increased from baseline in MNs after 24 h of MTZ application compared to those incubated in DMSO alone or those not expressing NTR (Supplemental 9G–I and Supplementary Fig. 12A, B). Similarly, $cacnb1^{-/-}$ mutants, which are immobile due to loss of skeletal muscle function but maintain normal MN activity[73], also showed increased GCaMP7 signal in response to MTZ treatment without tricaine immobilization (Supplementary Fig. 12A–C). Thus, GCaMP7 can be used as a GEDI and an accurate measure of neuronal death in immobilized zebrafish.

Although GCaMP is effective at labeling neuronal death within tricaine-anesthetized larvae, tricaine application can have adverse effects on physiology[74]. The dampening of neuronal activity within the zebrafish can potentially complicate the interpretation of neurodegenerative disease models, especially those in which hyperexcitability is thought to be a disease-associated phenotype, such as AD, FTD, and ALS[75,76]. Thus, a true GEDI would be preferable to GCaMP because it would eliminate the need for immobilization in zebrafish imaging preparations in which CNS activity is preserved, such as fictive swimming assays[72,73,77]. We first tested the ability of RGEDI-P2a-EGFP to detect neuronal death in vivo by co-injecting DNA encoding *neuroD*:NTR-BFP and *neuroD*:RGEDI-P2a-EGFP at the 1-cell stage, and then using in toto live longitudinal imaging to track fluorescence of co-expressing neurons within the larval spinal cord after NTR-MTZ–mediated ablation beginning at 72 hpf (Fig. 5A). After 24 h of incubation in 10 μM MTZ, the morphology of neurons showed signs of degeneration including neurite retraction and loss of fluorescence, yet neurons co-expressing NTR-BFP and RGEDI-P2a-EGFP did not show increases in GEDI ratio (Fig. 5B, C), indicating RGEDI signal cannot distinguish live from dead neurons in this system.

We hypothesized that extracellular $Ca^{2+}$ levels in vivo within the zebrafish larvae could be too low to reach the concentration required for RGEDI to optimally fluoresce. Therefore, we next tested if GC150, which has a higher binding affinity for $Ca^{2+}$ (Fig. 4E), could better report neuronal death in vivo. Sporadic expression of GC150-P2a-mApple was generated by co-injection of DNA encoding *neuroD*: GC150-P2a-mApple with *neuroD*:NTR-BFP at the 1-cell stage. Live longitudinal imaging was performed on larvae incubated in either DMSO or 10 μM MTZ, and individual neurons within the brain expressing both mApple and BFP were tracked in 4D within the whole larvae (Fig. 5D–F). Larvae exposed to 10 μM MTZ showed increased GC150 signal by 24 h after MTZ extending to 48 h, while larvae exposed to DMSO alone did not show signs of neuronal death or increases in GC150 (Fig. 5D–F). With increased binding affinity compared to RGEDI, GC150 could potentially be more susceptible to detecting physiological $Ca^{2+}$ transients within neurons, similar to GCaMP, which could confound its utility as a GEDI. To test whether GC150 increases in fluorescence during $Ca^{2+}$ transients, GC150 was targeted to MNs by injection of *mnx1*:GC150-P2a-mApple. Larvae were immobilized in BTS, and no response of GC150 was detected after activation of the motor circuit, indicating GC150 signal does not increase in response to normal calcium transients within neurons (Supplementary

Fig. 11F, G). These data indicate GC150 is suitable for in vivo detection of neuronal death in an un-anesthetized animal.

## Discussion

The study of neurodegenerative diseases has been hampered by an inability to distinguish populations of neurons destined to die from those that have already perished, which precludes the investigation of mechanisms that drive selective degeneration. Here, we characterize a biosensor family, GEDI, which is specifically tuned to detect neuronal death in longitudinal imaging studies, facilitating analysis of neurons in time points leading up to neuronal death. Using automated microscopy, we show that the analysis of GEDI is compatible with fully automated, single-cell survival analysis, which has been previously shown to be 100–1000 times more sensitive than population-based studies that rely on a single snapshot in time[15]. We believe this tool will lead to more precision in the discovery of the mechanisms of neurodegeneration and increase the throughput of quantitative studies to discover novel therapeutics.

Similar to other death-pathway agnostic indicators of cell death or viability, GEDI detects the loss of membrane integrity as a readout of cell death[78]. However, by uniquely measuring the $Ca^{2+}$ permeability of the membrane, GEDI holds several key advantages for longitudinal imaging. For one, unlike death indicators that rely on DNA intercalation, such as EthD1, PI, or 4′,6-diamidino-2-phenylindole (DAPI)/Höechst, GEDIs are nontoxic. Additionally, intercalating agents require a breach of both the plasma membrane and the nuclear membrane, which can be problematic in the stochastic process of degeneration and could result in delayed signal (Supplementary Fig. 2). In contrast, each indicator we used to engineer GEDIs binds $Ca^{2+}$ in less than a second[36,37,79], which is over 500× faster than the time course of death after $NaN_3$ exposure (Fig. 4). GEDIs can also be combined with complementary labels specific for cell death pathways (Supplementary Figs. 2, 3), which could help resolve ambiguity in cell death pathway crosstalk such as when a cell resorts to a different cell death pathway when the primary pathway is blocked[21], by providing more direct temporal linkage between death pathway signal and death. The stability of the fluorescence tags used in the GEDIs (Fig. 2D) enables the signal to be sampled at long time intervals such as every 24 h, a particularly important property during long-term imaging studies to minimize phototoxicity. These properties allowed us to use GEDIs to empirically determine the level of cytosolic $Ca^{2+}$ associated with an irreversible fate (Fig. 1 and Supplementary Fig. 4), a property no other death indicator is capable of, to our knowledge. Furthermore, with the increased time resolution of death possible using GEDI, subpopulations of death-resistant neurons and the spread of neuronal death can now be imaged. For instance, reports of subpopulations of neurons displaying resistance to glutamate have been described in culture and in vivo[80–83], yet each report has relied on dyes to characterize death at static time points, limiting the ability to resolve the time course and cell-to-cell transmission of excitotoxic injury on a single-cell level now possible with GEDI (Fig. 2A–D). While this study focused on the use of GEDI in degenerating neurons, it should be noted that most if not all cells maintain a concentration gradient of $Ca^{2+}$, suggesting that these death indicators could be adapted for use to report cell death and spread of cell death in other cell or tissue types[84].

Although much of what we know about the etiology of neurodegenerative disease has come from a two-dimensional culture of neurons, it is becoming increasingly clear that the progression of neurodegenerative disease is dependent on the relationship of neurons to their surrounding tissue. For

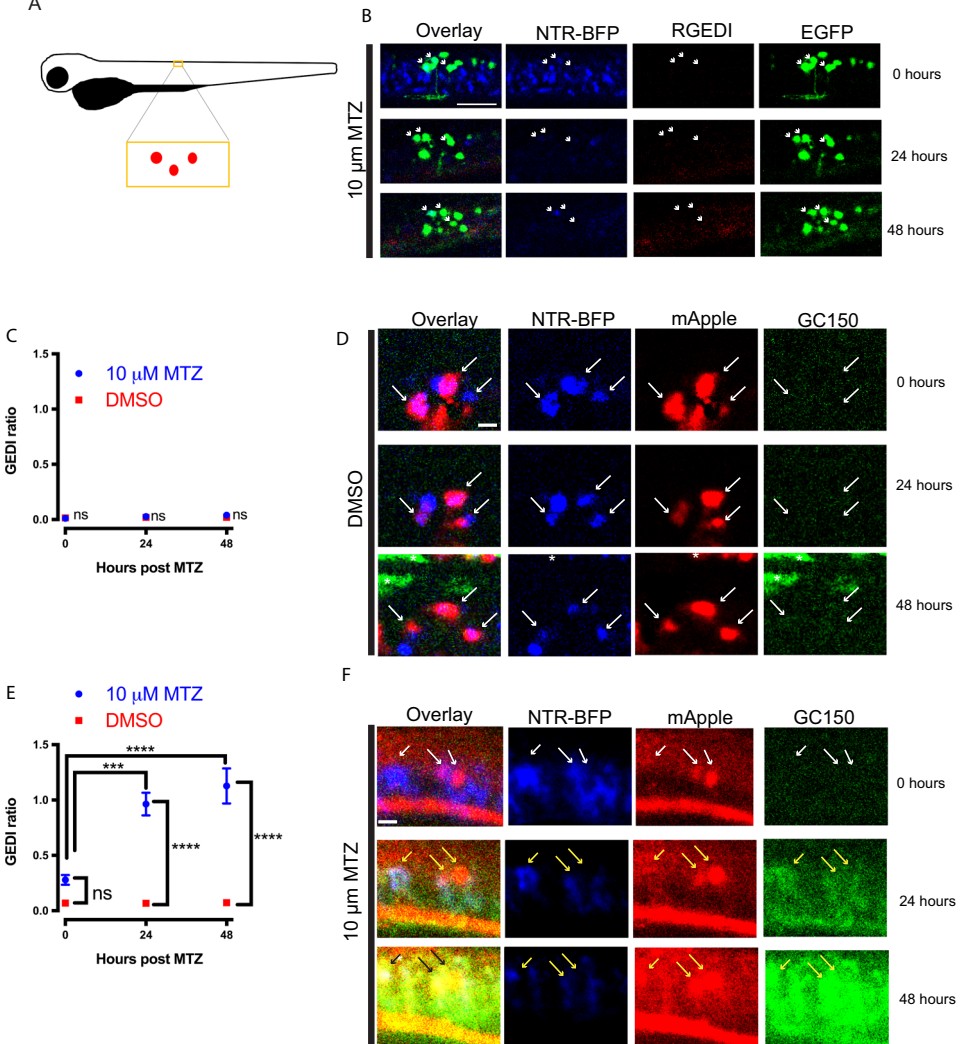

**Fig. 5 Single-cell tracking and specific detection of death within live zebrafish larvae with GC150 but not RGEDI. A** Cartoon schematic of zebrafish larvae showing the approximate location of sparsely labeled clusters of neurons. **B** Neurons in the zebrafish larvae spinal cord co-expressing NTR-BFP, EGFP, and RGEDI at 0, 24, and 48 h after mounting for automated imaging in MTZ. White arrows indicate neurons co-expressing NTR-BFP and EGFP. Scale = 50 μm. **C** Quantification of GEDI ratio in neurons co-expressing NTR-BFP and RGEDI-P2a-EGFP exposed to MTZ ($n = 7$) or DMSO ($n = 5$) showing no increase in GEDI signal (ANOVA Sidak's, ns not significant). **D** Neurons in the zebrafish larvae spinal cord co-expressing NTR-BFP, mApple, and GC150 at 0, 24, and 48 h after mounting for automated imaging in DMSO. White arrows indicate neurons with the fluorescence of BFP and mApple, asterisks indicate autofluorescence from pigment in larvae skin, scale = 20 μm. **E** Quantification of average GEDI ratio of neurons from zebrafish larvae incubated in DMSO ($n = 7$) or 10 μM MTZ ($n = 17$) over time showing an increase in GEDI ratio in neurons indicating neuronal death after 24 h in MTZ (ANOVA Tukey's ****$p < 0.0001$, ***$p < 0.0005$). **F** Neurons in zebrafish larvae spinal cord co-expressing NTR-BFP, mApple, and GC150 at 0, 24, and 48 h after mounting for automated imaging in 10 μM MTZ. White arrows indicate neurons with the fluorescence of BFP and mApple, yellow arrows indicate neurons with the fluorescence of BFP, mApple, and GC150, indicating neuronal death. Asterisks indicate autofluorescence from pigment in larvae skin, scale = 20 μm. Error bars represent SEM. The experiment was on 12 larvae with similar results.

instance, multiple neurodegenerative diseases are linked to abnormal circuitry[75,76,85,86], and evidence suggests that propagation of neurodegenerative disease throughout the brain may proceed via the spread of pathogenic proteins[87–89]. To study such cell nonautonomous phenomena of neurodegeneration, an approach that integrates the three-dimensionality of tissue is required. Previously, we used organotypic slice culture to create a more tissue-like environment in which to study neurodegeneration over time, with an automated 4D imaging platform[90]. Here, in combination with GEDI, we apply this technology to a zebrafish model of neurodegeneration to study neuronal death in vivo. In contrast to organotypic slice culture models, which expose neurons to the stresses of brain dissection and culturing, in vivo 4D imaging of zebrafish larvae fully

preserves the architecture of the brain. Zebrafish larvae studies can also be scaled up for high-throughput imaging screens[91], and to our knowledge, our platform represents the first optimized for 4D longitudinal imaging of immobilized fish. This can be especially useful, as zebrafish larvae are a well-characterized behavioral model[92,93], and 4D imaging can be used in parallel with behavioral analysis, providing an important behavioral correlate of neurodegeneration over time.

Our use of 4D imaging of zebrafish generated several important findings. First, we show that the loss of fluorescence from neurons expressing mCherry is not an acute indicator of neuronal death in vivo (Supplementary Fig. 9). One implication of this finding is that studies of neurodegeneration using fluorescent proteins in zebrafish could underestimate the time course of

degeneration in the model, and behavioral characterization may be a more acute indicator of degeneration. Time-lapse imaging of zebrafish neurodegeneration is also complicated by the species' high neuronal regeneration capacity[94] and the presence of scavenger cells such as microglia[62], which could alter the apparent rate of death overtime, underscoring the necessity of an acute indicator to track neurons as they die over time. Second, our successful application of GCaMP7 to detect neuronal death in tricaine-immobilized zebrafish larvae means that commonly used GECIs, under neuronal paralytic conditions, can act as GEDIs. While GECIs prove to be a convenient tool for the study of neurodegeneration, this finding should also raise caution in the interpretation of GECI systems under conditions in which neuronal death occurs, as a chronic increase in GECI signal after death could be confused with normal $Ca^{2+}$ transient activity. Finally, we showed that although the RGEDI construct could not be used to detect death in vivo, GC150 and GCaMP7 could. Due to the differences in $Ca^{2+}$ binding affinity between the three indicators, these data suggest that free $Ca^{2+}$ in the extracellular spaces in the brain of the zebrafish larvae is somewhat limited and may be insufficient to induce the RGEDI fluorescence upon cell death. Thus, the difference in the functionality of RGEDI in cultured neurons (Figs. 1–4) and in vivo in the zebrafish brain (Fig. 5 and Supplementary Fig. 9), could be due to the virtually unlimited supply of $Ca^{2+}$ in culture medium compared to the brain, where extracellular $Ca^{2+}$ can be limiting in times of high activity[95] and in dense synaptic areas[96]. Interestingly, the limited extracellular $Ca^{2+}$ in our in vivo assays raises the intriguing possibility that the sequestration of free calcium in dying neurons could be a previously unexplored mechanism of cognitive decline in neurodegeneration. Future studies using 4D modeling will be key to address this question.

With few disease-modifying therapies for neurodegenerative diseases available, there is a great need to understand disease mechanisms and etiology to develop new therapeutic targets[12]. Our studies using neurodegenerative disease models of PD, ALS/FTD, HD, glutamate toxicity, and in vivo neuronal ablation demonstrate the ability of GEDIs to acutely identify the moment of death in time-lapse imaging studies, allowing a unique time-resolved view of neurodegeneration. We believe the use of GEDIs will aid longitudinal single-cell analysis of neurons to complete our understanding of the underlying causes of neurodegeneration, and provide assays to help generate much-needed therapeutics.

## Methods

**Animals and culturing**. All animal experiments complied with UCSF Institutional Animal Care and Use Committee (IACUC, protocols AN183829-02 and AN189188-01) regulations. Animals are housed in approved facilities with humidity regulated between 30 and 70%, temperature between 68 and 79 °F, and 12 h light/dark cycles. Primary mouse (C57BL/6) and rat (Long-Evans) cortical neurons were prepared at embryonic days 20–21. Brain cortices were dissected in dissociation medium (DM-81.8 mM Na2SO4, 30 mM K2SO4, 5.8 mM MgCl2, 0.25 mM CaCl2, 1 mM HEPES, 20 mM glucose, 0.001% phenol red, and 0.16 mM NaOH) with kynurenic acid (1 mM final) (DM/KY). KY solution was diluted from stock 10x KY solution (10 mM KY, 0.0025% phenol red, 5 mM HEPES, and 100 mM MgCl2). For cell disassociation, the cortices were treated with papain (100 U, Worthington Biochemical) for 10 min, followed by treatment with trypsin inhibitor solution (15 mg/mL trypsin inhibitor, Sigma) for 10 min. Both solutions were made up in DM/KY, sterile filtered, and kept in a 37 °C water bath. The cortices were then gently triturated to dissociate single neurons in Opti-MEM (Thermo Fisher Scientific) and glucose medium (20 mM). Neurons were plated into 96-well plates at a density of 100,000 cells/mL. Two hours after plating, the plating medium was replaced with a Neurobasal growth medium with 100X GlutaMAX, pen/strep, and B27 supplement (NB medium). Zebrafish embryos raised to 48 hpf were enzymatically dechorionated using 2 mg/ml Pronase (Protease, Type XIV, Sigma) for 20 min. For behavioral analysis, embryos at 72 and 96 hpf were lightly tapped on the tail with #2 forceps while recording a 10 s movie and analyzed for movement within the dish by binarizing the image and quantifying

movement[97]. A list of all zebrafish lines used in this study are available in Supplementary Table 2.

**Plasmids, transfections, toxins, dyes, injections, and transgenics**. The mammalian expression constructs phSyn1:RGEDI-P2a-EGFP, phSyn1:RGEDI-P2a-3xTagBFP2, phSyn1:RGEDI-NLS-P2a-EGFP-NLS, and phSyn1:TDP43, phSyn1:empty were generated by synthesizing the insert into a pBluescript Sk + backbone. All constructs were verified by sequencing. At 4–5 days in vitro (DIV), rat cortical neurons were transfected with plasmids and Lipofectamine 2000 to achieve sparse labeling of neurons within each well. For survival analysis, each well of a 96-well plate containing primary rat cortical neurons was cotransfected with 0.15 µg of DNA of phSyn1:RGEDI-P2a-EGFP, phSyn1:RGEDI-P2a-3xTagBFP2, phSyn1:R-GEDI-NLS-P2a-EGFP-NLS, and phSyn1:mRuby-P2a- GCaMP6[58], 0.1 µg of DNA of phsyn1:empty, pGW1-HttEx1Q97-mCerulean, pGW1- HttEx1Q25-mCerulean[3], or 0.075 µg of DNA pCAGGS–α-synuclein[39] or phSyn1:TDP43. L-Glutamic acid monosodium salt was diluted in NB media with 0.5% DMSO. 2% NaN3 (Sigma) was dissolved in NB media, or in PBS with or without Ca2+. For TUNEL staining, the Alexa647 Click-iT Assay for in situ apoptosis detection was used (Life Technologies). For Caspase detection, PhiPhiLux (Gentaur) and NucView 405 Caspase Substrate (Biotium) were used according to directions[23,68,98]. For testing other cell death indicators, mouse primary cortical neurons were isolated from embryonic 17-day pups and at 3 DIV, neurons were transfected with either 0.02 ug of hSyn1:RGEDI-P2a-EGFP or pGW1-EGFP. At 6 DIV, neurons transfected with RGEDI were treated with neural basal media while neurons transfected with EGFP were treated with 1 uM ethidium homodimer-1 Life Technologies) or 0.5 uM propidium iodide (Life Technologies). Dyes were allowed to incubate for 30 min before a pretreatment timepoint was taken. Neurons were exposed to 90 s UV light with a custom-built LED light box to induce cell death and cells were imaged every 4 h to track the signal of death indicators.

For experiments using electrical field stimulation, primary mouse neurons were cultured for 4 days before transfection with phSyn1:RGEDI-P2a-EGFP, phSyn1:mRuby-P2a-GCaMP6f, or phSyn1:GC150-P2a-mApple and then imaged at 8 DIV in Tyrode's medium (pH 7.4; in mm: 127 NaCl, 10 HEPES-NaOH, 2.5 KCl, 2 MgCl2, and 2 CaCl2, 30 mm glucose and 10 mm pyruvate) using a Nikon CFI Plan Apo ×40/0.95 numerical aperture air objective on a Nikon Ti-E inverted microscope with an iXon EMCCD camera (Andor Technology). Field stimulation (3 s at 30 Hz) was done using an A385 current isolator and an SYS-A310 Accupulser signal generator (World Precision Instruments). NaN3 was then directly injected into the imaging chamber to achieve rapid mixing. GCaMP6f and RGEDI fluorescence images were obtained using the following filters [490/20(ex),535/50 (em) and 543/22 (ex), 617/73 (em) respectively, Chroma] and regions of interest were drawn over cell bodies. The fold change (ΔF/F0) in fluorescence was calculated for each time point after background subtraction.

To generate zebrafish constructs, gateway recombination-based cloning was performed (Life Technologies) using the Tol2kit[99]. A pME entry clone was generated for RGEDI-P2a3xBFP and RGEDI-P2a-EGFP by subcloning from phSyn1:RGEDI-P2a3xBFP, phSyn1:RGEDI-P2a-EGFP, or phSyn1:GCaMP6f-P2a-mRuby using the pCR8/GW/TOPO TA cloning kit (Life Technologies). pME:GC150-P2a-mApple and pME-NTR-BFP were synthesized (Genscript) and cloned into pET-30a. Each was combined with p5E-neuroD[100], P3E polyA and pDestTol2pA[99], to generate neuroD: RGEDI-P2a-EGFP, neuroD: GC150-P2a-mApple, and neuroD:NTR-BFP. Embryos were injected at the 1-cell stage into the yolk with ~20 nl containing 100 ng/ul of each DNA. For stable transgenic neuroD:GC150-P2a-mApple, embryos were co-injected with neuroD:GC150-P2a-mApple and T7 in vitro transcribed tol2 mRNA (100 ng/ul) (Bio-Synthesis, Inc) into the cell at the 1-cell stage, and then raised to adulthood. Progeny were then screened by outcrossing to identify founder lines that were propagated and used for subsequent experiments. A list of all constructs used in this study are available in Supplementary Table 3.

**In toto automated 4D high-content imaging of immobilized zebrafish**. For imaging experiments, zebrafish embryos were dechorionated at 24 hpf, sorted for fluorescence, and put into 200 µM 1-phenyl 2-thiourea (PTU) E2 medium at 28.5 °C to inhibit melanogenesis[101]. At 72 hpf embryos were immobilized in 0.05% tricaine methanesulfonate (Sigma) or 50 µM BTS (Tocris) for an hour, then were loaded into wells of a ZFplate (Diagnocine) in 100 ul of media with paralytic using wide-bore tips and allowed to settle into the slits. Using a multichannel pipette, 100 ul of molten 1.5% low melting point agarose was loaded into each nozzle hole, and then 100 ul of agarose/paralytic mixture was immediately removed and agarose was allowed to solidify. Finally, liquid media was added to the top of each well containing paralytic with DMSO and/or with 10 uM MTZ (Sigma).

The automated spinning disk confocal imaging system was previously described[25]. Briefly, a custom system was used combining a Nikon microscope base (Nikon Ti-E), a Yokogawa spinning disk confocal (Yokogawa CSU-W1), an automated three-axis stage (Applied Scientific Instrumentation, MS-2500-Ti and PZ-2300), and modified custom software controlling Micromanager (version 2.0 gamma) allowing an automated return to the same location on the imaging plate for continual imaging of the same location in 3D space. Automation of the system was performed with Green Button Go (Biosero, Freemont). To accommodate zebrafish imaging at 28.5 °C rather than at the enclosure temperature

of 37 °C for mammalian cell culture, a custom-made homeostatic Peltier cooling lid was designed and constructed (Physiotemp) to sit on top of the ZF plate.

**Data analysis and quantification.** Quantification of GEDI and morphology channel fluorescence intensity from 2D cultures was done using files obtained by automated imaging as previously described[3,14,102]. Files were processed using custom scripts running within a custom-built image processing Galaxy bioinformatics cluster[25,103] that background subtracts, montages, fine-tunes alignment across time points of imaging, segments individual neurons, tracks segmented neurons over time, and then extracts intensity and feature information from each neuron into a csv file. Background subtraction was performed by subtracting the median intensity of each image and was required for the calculation of a translatable GEDI signal across data sets. Segmentation of neurons was targeted towards detection of the brightest area of the morphology of the neuron, usually the soma, that was larger than a minimal size threshold of 100 pixels. The GEDI ratio is sensitive to extent of segmentation of neurons, and errors in segmentation that include background can reduce the precision of the GEDI ratio obtained. A tight segmentation of neurons around the soma is desirable as segmentation of neurites frequently results in segmentation of multiple neurons at a time because of overlapping projections, and the dimmer and inconsistent morphology fluorescence signal extension throughout neurites. Tracking of neurons was performed by labeling an object as the same object at the next time point based on the proximity of the coordinates of the segmented mask to a segmented mask at the previous time point. Survival analysis was performed by defining the time of death as the point at which the GEDI ratio of a longitudinally imaged neuron exceeds the empirically calculated GEDI threshold, or the point at which point a tracked segmentation label is lost, which was performed using custom scripts written in R. The GEDI threshold was determined using the following equation:

$$\text{GEDI ratio threshold} = ([(\text{mean GEDI ratio dead}) - (\text{mean GEDI ratio live})]*0.25) \\ + [\text{mean GEDI ratio live}]$$

(1)

The survival package for R statistical software was used to construct Kaplan–Meier curves from the survival data based on the GEDI ratio, and survival functions were fit to these curves to derive cumulative survival and risk-of-death curves that describe the instantaneous risk of death for individual neurons as previously described[38]. Linear regressions of log decay and nonlinear regressions of GEDI signal increase were calculated in Prism using plateau followed by one-phase association kinetics.

For zebrafish motor axon area quantification, in toto z-stacks of immobilized fish were maximum projected, stitched together, background subtracted, and binarized. The spinal cord soma, brain, and eye fluorescence typical of *mnx1* transgenics[66] was manually masked out using FIJI (version 1.53c), leaving only the motor axon projections and the total area of the signal was quantified per fish and standardized to the initial time point for that fish. For zebrafish analysis of GEDI signal, in toto z-stacks of immobilized fish were stitched together, background subtracted per z plane, and all imaging channels over time were combined into a 5D composite hyperstacks (x, y, z, color, and time dimensions). Due to a spherical aberration commonly present in confocal 3D imaging and present on this platform[25,104], maximum projections across z planes created blurry images that limited precision in quantification, so individual z planes were used for all fluorescence quantifications of GCaMP 7 and GEDI. In Tg:mnx1:GCaMP7/NTR-mCherry experiments, GEDI ratio was calculated per hemi-segment because individual neurons could not be easily resolved. In neuroD:NTR-BFP experiments, individual neurons were located by the co-expression of the GEDI morphology channel with the NTR-BFP while blinded to the GEDI fluorescence.

**iPSC differentiation to MNs.** iPSC line derived from a healthy control individual (KW4) was obtained from the Yamanaka lab[96]. A line containing the SOD1 D90A mutation was acquired from the iPSC repository at the Packard Center at Johns Hopkins. Reprogramming and characterization of the SOD1 D90 iPSC line were previously reported[105]. Healthy and SOD1 D90A iPSCs were found to be karyotypically normal and were differentiated into MNs using a modified dual-SMAD inhibition protocol[106] (http://neurolincs.org/pdf/diMN-protocol.pdf). At day 18 of differentiation, iPSC-derived MNs were dissociated using trypsin (Thermo Fisher), embedded in diluted Matrigel (Corning) to limit cell motility, and plated onto Matrigel-coated 96-well plates. From day 20–35, the neurons underwent a medium change every 2–3 days.

**Reporting Summary.** Further information on research design is available in the Nature Research Reporting Summary linked to this article.

## Data availability

Raw data acquired by robotic microscopy used in this study are too large to post online but the raw and processed data that support the findings of this study can be obtained from the corresponding author upon reasonable request. Additional representative data, measurements, and analysis scripts are available at: https://doi.org/10.5281/zenodo.5107973.

## Code availability

The code is copyright protected, and its use in performing the described research is patented (US Patent 7,139,415 and US Patent Application 14/737,325). Code is available upon request to the corresponding author. Access to and use of the code is subject to a nonexclusive, revocable, nontransferable, and limited right to use the code for the exclusive purpose of undertaking academic, governmental, or not-for-profit research. Use of the code or any part thereof for commercial or clinical purposes is strictly prohibited in the absence of a Commercial License Agreement from The J. David Gladstone Institutes.

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

## Acknowledgements

This work was supported by grants from the NIH (U54 NS191046, NS101996, RF1 AG058476, RF1 AG056151, RF1 AG058447, P01 AG054407, U01 MH115747, R01 LM013617 and RF1 AG064170), as well as support from the Koret Foundation Artificial Intelligence Program for Biomedical Research and the Taube/Koret Center for Neuro-degenerative Disease Research (SF). The Gladstone Institutes received support from a National Center for Research Resources Grant RR18928. Z.D. was supported by a Hillblom Graduate Fellowship. We gratefully acknowledge the support of NVIDIA Corporation with the donation of the Titan Xp GPU used for this research. Kathryn Claiborn provided editorial assistance, Kelley Nelson and Gayane Abramova administrative assistance, Caitlyn Bonilla helped troubleshoot the microscope, and David Cahill provided lab maintenance and organization. Elliot Mount provided key assistance in configuring and troubleshooting the microscope to image live zebrafish. Mnx1 plasmid and transgenic zebrafish line were a kind gift from E. Isakoff (UC-Berkeley). Jack Taylor, Matthew Mccarroll, Louie Ramos, and Ethan Fertsch (UCSF) and Claire Quinata (Gladstone Institutes) provided zebrafish expertise and colony maintenance. Sean Low (University of Michigan) provided zebrafish expertise, mentoring, guidance, and calcium imaging assistance.

## Author contributions

J.W.L. and S.F. wrote the manuscript. J.W.L., K.S., N.C., K.N., A.J., and S.F. designed the automatic microscopy experiments. Molecular biology, biosensor design, and rodent primary neuron culturing and imaging performed by J.W.L. J.W.L. and D.K. designed zebrafish experiments. J.W.L., V.O., and W.L. performed zebrafish microinjections and imaging experiments, J.W.L. and J.M. performed zebrafish behavior experiments. N.C., M.C., S.W., D.H., and Z.D. carried out death delay imaging experiments with stimulation, dyes, stains, and GEDI. K.S. performed iPS differentiation, culturing, transfection, and imaging experiments. Custom scripts for analysis of imaging experiments done by J.L. and K.S. All authors reviewed the manuscript.

## Competing interests

The authors declare no competing interests, but the following competing financial interests: S.F. is the inventor of Robotic Microscopy Systems, US Patent 7,139,415 and Automated Robotic Microscopy Systems, US Patent Application 14/737,325, both assigned to the J. David Gladstone Institutes. A provisional US and EPO patent for the GEDI biosensor (inventors J.W.L., K.S., and S.F.) assigned to the J. David Gladstone Institutes has been placed GL2016-815, May 2019.
