## [Peer Review File · Nature Communications]

Genetically encoded cell-death indicators (GEDIs) to detect an early irreversible commitment to neurodegenerationReviewers' comments:

Reviewer #1 (Remarks to the Author):

In this work, the authors have developed a new assay for the detection of early cell death events (apoptosis, necrosis, etc.) in vivo.

To this end, the authors decided to rely on changes in cytosolic calcium levels as a marker.

Although partly questionable (this parameter is not necessarily adequate for all types of cell death), it is an interesting idea that deserves experimental validation.

First, the authors implement a series of in cellulo experiments showing convincingly that the indicator system they developed is capable of detecting calcium variations resulting from various death-inducing stimuli in nerve cells (primary neurons).

Second, the authors make use of the zebrafish model to validate their system in vivo. Overall the results are interesting, however some issues remain to be clarified at this stage.

Below are some specific points that may help improve the manuscript.

Fig. 6 D,F. As far as mApple detection is concerned, the signal-to-noise ratio is very low in MTZ panels compared to control panels (DMSO), contrary to BFP panels. Actually it seems that, for some reason, the mApple/GC150 expression vector did not really work in MTZ experiments (Fig 6F), resulting in a weak GC150 signal. These are key experiments, and as such they really deserve to be optimized in terms of signal-to-noise ratio.

Fig. S6. Showing only merged images does not facilitate understanding. Regarding the A/24h panel, it is unclear why the mCherry signal has almost disappeared. It seems that the expression vector does not really work in A panels (MTZ) compared to B panels (Tricaine), including at time 0h, which undermines observations.

In panel A/24h, the mCherry signal drop suggests that a significant number of cells are actually already dead. Measuring caspase activity at 24 hours may therefore be too late. A longitudinal kinetic study should help clarify this issue.

In this respect, it would be useful to carry out comparable kinetics regarding GC150 (Figure 6F) in order to have an idea of the time course of caspase 3 activation in connection with the cytosolic calcium peak.

In the same line, and in order to further validate this new cell death detection system, it is important to compare the kinetics of the GC150 signal with proven cell death markers such as TUNEL assay, which can be implemented in zebrafish.

Minor points

Fig. 3B, F: This kind of representation is somewhat difficult to understand. Histograms might be easier to handle.

Fig. 5C, I: The use of red and green is confusing for the reader since they may incorrectly refer to fluorescent markers (see Fig. 5G).

Reviewer #2 (Remarks to the Author):

Linsley et al. used a novel calcium indicator in the cytoplasm to visualize cell death prior to other signs such as morphological changes or traditional indicators. They examined biologically relevant conditions using primary culture, iPSCs, and in vivo model.

The idea is innovative and has significant impact on the neuroscience research. Experimental designs and analyses are solid, and the logic is easy to follow. I have several concerns/questions listed below.

Fig.1J Separation of untreated vs +NaN₃ populations is clear. The variability of GEDI ratio in the NaN₃ treated population seems bigger compared to that of in vivo system (for example, Fig.5C or I). Is this also due to the higher extracellular Ca concentration ?

Related to the question above, is there any literature that Ca concentration of the extracellular fluid in zebrafish is significantly lower than 2 mM used in the culture experiments?

#p10 L2 Fig 2C should be Fig2D.

#Fig2E How does the decrease of fluorescence compare to the disintegration of the plasma membrane, with regards to the time course? In other words, how much of the decrease is caused by the active degradation as opposed to the simple diffusion into the extracellular space?

#Fig.3G Neurons showed lower survival rate in the ~200 hr culture period. Given the time course of the disease in human patients, the rapid change observed in the experiment is somewhat unexpected. Unlike the experiments in Fig.3A-C, genetic factors causing the disease is expected to be similar to those in patients from whom iPSCs are derived. Do authors have any idea to explain this difference?

#Fig.4D In the second paragraph of p13, authors state that "As expected cells expressing GC150-P2a-mApple showed increased GEDI ration." Is GC150-P2a-mApple compared to RGEDI-P2a-3XBFP here? I am confused because the GECI and the reference are both changed. Please clarify.

P15 I3 " mnx:Gal4; UAS:mCherry; UAS:EGFP zebrafish larvae " seems to be a mistake. It should contain NTR under UAS.

#p16 I3 GcaMP7 needs to be in capital letters.

#Fig.5B-D The immobility of 24 hr larvae in the absence of Ca increase is highly unexpected and important. Authors discuss the implication of this finding on p20. Please expand this discussion so as to include possible mechanisms leading to the axon retraction in the absence of Ca increase.

Reviewer #3 (Remarks to the Author):

This manuscript by Linsley and colleagues describes the generation of novel "Genetically Encoded Death Indicators (GEDIs)" that show increased fluorescence with high concentrations of Ca. The idea of developing such death indicators is elegant and is based on generating variations of previous Ca indicators. The GCAMP6 Ca indicators can detect transient increases in Ca, and these have been used widely to study neuronal activity. Recently, variants of GCAMP6 that have a higher Kd for Ca have been developed with ER retention signals. These have allowed investigators to detect Ca in the ER (where Ca concentration is high) without detecting the smaller Ca transients that occur during neuronal activity. The authors have now used these high Kd Ca indicators and removed the ER retention signals. This simple manipulation allows them to detect Ca in the cytoplasm- but only when Ca levels are substantially increased- such as when the plasma membrane integrity is lost and extracellular Ca gets inside the cells.

The authors have done a very thorough job of characterizing these GEDIs and show convincingly that they can detect dying cells in a highly automated process. Importantly, they have demonstrated this in different contexts where neurons degenerate (e.g. glutamate excitotoxicity, expression of synuclein, mutant HTT, and TDP43), including iPS cell derived motor neurons from patients carrying a mutation in SOD. These are indeed an impressive array of contexts. The authors have then generated variants of the GEDIs (blue fluorescence, fluorescence with NLS, a version based on GCAMP6-150 with higher affinity to Ca) and shown their functionality. Lastly, the authors have utilized the zebrafish model where they use their high sophisticated imaging techniques to show the feasibility of using GCAMP6 as a death indicator in immobilized fish (where neuronal firing is blocked) as well as the GCAMP6-150 based GEDI under non-immobilized conditions.

Overall, this is a very thorough study showing the functionality of various GEDIs as cell death indicators. While the GEDI essentially detect membrane permeabilization, which is also what

several previous death indicators detect (e.g. PI staining), the fact that these are genetically encoded, make them much more attractive tools for longitudinal in vivo and in vitro studies. This manuscript is essentially a tool development-focused manuscript that demonstrates the capability of these new death indicators.

Comments:

1) On p7, the authors indicate that the increase in GEDI ratio preceded obvious morphology changes. However, Fig. 1H does not show that clearly. Either a better image is needed here or the authors should clarify this statement.

2) Figure 5 showing the capability of using GCAMP7 as a death indicator – but only under conditions when the zebrafish are immobilized seems a bit out of place given the focus of the rest of the manuscript on these new GEDIs. It seems highly unlikely that GCAMP7 will be used as a death indicator because of the need to block neuronal activity. There is value in making this point and perhaps the authors could consider moving this figure to the supplement.

3) The authors state in the abstract that the GEDIs detect Ca levels that cells achieve early in their death process. However, as shown by the authors, the GEDIs detect Ca that enters the cell after the plasma membrane integrity is lost. Therefore, these indicators are essentially detecting loss of plasma membrane integrity, which would not be considered an early event in the cell death process. The language here would need to be corrected.

Reviewer #4 (Remarks to the Author):

In this manuscript, Linsley and colleagues demonstrated application of genetically encoded Ca²⁺ indicators GECIs to visualize cell-death. In order to detect pathway independent total cell-death, low affinity GECIs were used to visualize the intracellular Ca²⁺ increase due to the permeabilization through the disrupted membrane. They demonstrated detection of cell-death specific signal change in rat cortical primary neurons with P2A peptide-based cotransfection of CEPIA and EGFP. By the ratiometric analysis of fluorescent probes, they succeeded in the detection of early stage of cell-death induced from different pathways. They also detected cell death of Neurons derived from iPSC similarly. Since CEPIA and EGFP combination failed to in vivo detection in motor neurons of zebrafish larvae, GCaMP6-150 and mApple was used as an alternative. Although the authors claimed development and application of a GEDI, reality of the “GEDI” is an already reported GECI “CEPIA” itself and versatility of the detection for different cell types is questioned. Overall, this manuscript holds difficulty for publication from this journal, because it does not adequately represent the perceived importance of the technology.

Major points:

1. The authors developed GEDI from RCEPIA by removing the ER retention signals. However, the following explanation in the original paper, “We expressed the colour variants of CEPIA with ER-targeting signal sequences in HeLa cells”, indicates that “CEPIA” stands for the variants without target signal sequence. To avoid confusion, different names CEPIA and GEDI should not be used for the same indicator. The term “GEDI” should be used to refer to a broad range of genetically encoded death indicator including GCaMP6-150, but not to refer to RCEPIA. The title of the manuscript must be also toned down to reflect this work is one of the applications of the low affinity GECI.

2. Intensity of EGFP and RGEDI seems saturated at the high intensity region in the Figure 1H. If the saturated values are used for the calculation of the graph in Figure 1I, the graph contains significant error. In addition, intensity ratio image instead of the overlay image is preferable to compare with the graph in Figure 1I.

3. Since the contents of the section of the “GCaMP acutely reports death in vivo” is out of the concept of this manuscript, “dead-cell detection by low affinity GECI”, it should be removed from

the main text and described in the supplementary note if the authors want. Moreover, the indicator for the zebrafish specific artificial condition is not so much attractive for a broad range of readers who are interested in cell-death imaging.

4. Different sensitivity of GFP and RFP to lysosomal proteases is mentioned as a key factor for developing tandem tag biosensors to measure autophagy and Reference-46 is cited for supporting that (p. 10 L8). However, according to this paper, the sensor was developed based on the "different pH stability of EGFP and mRFP", and the word "protease" could not be found by my word search for this paper.

5. Although the relative dimness of BFP is mentioned as a reason of the smaller dynamic range of RGEDIP2a-3xBFP (p. 13 L4), the intensity difference of the reference fluorescence does not cause a difference in the dynamic range in principle. A more satisfactory explanation is required.

6. Response of the GC150-P2a-mApple to the field stimulation should also be checked to confirm whether its signal change is truly cell-death specific.

Minor points:

1. The frequency of the field stimulation in Figure 1B was described as 10 Hz (p. 6 L1 from the bottom) in the main text. However, it was 30 Hz in the Method section (p. 23 L7 from the bottom) and the figure legend (p. 27 L1 from the bottom). This discrepancy should be corrected.

2. Since NaN₃ has already been used in the description for the earlier experiment, an explanation about NaN₃, "a known cytotoxin", (p. 8 L4) might be unnecessary.

3. At p.11 L5-6 from the bottom, the authors mentioned "a clear separation of live and dead neurons could still be observed" for Figure 3B. However, there is no other way than the reference line drawn by manual curation to discriminate live and dead neurons in this figure. Clear discrimination based on the data is required. This is also true of Figure 3F.

4. The authors compared the rate of signal increase after exposure to NaN₃ between RGEDIP2a-3⁺BFP and RGEDIP2a-EGFP or GCaMP6f-P2a-mRuby58. However, it is difficult to compare the rates from Figure 4D. Independent graph, which can directly show statistical significance, is required.

5. The graph to show kinetics of the responses of GEDI (τ value), which can directly show statistical significance, must support description at p.13 L2-4 from the bottom.

6. "Figure 5D" at p. 15 L6 might be "Figure 5D-F".

7. At p. 20 L9, loss of what fluorescence does the authors mean?

8. At p. 23 L8 from the bottom, what the value 0.95 stand for? Numerical aperture?

9. In Figure 1J, although there are blue plots in "0 Minutes", no explanation for that in the figure legend. Green color might be correct.

Authors' Response to Reviewers' Comments

We thank the reviewers for their careful reading of our manuscript. Please find below a point-by-point response to the concerns raised during the initial review. The manuscript has now been thoroughly revised accordingly, and we hope that it is now acceptable for publication.

Reviewer #1:

Fig. 6 D,F. As far as mApple detection is concerned, the signal-to-noise ratio is very low in MTZ panels compared to control panels (DMSO), contrary to BFP panels. Actually it seems that, for some reason, the mApple/GC150 expression vector did not really work in MTZ experiments (Fig 6F), resulting in a weak GC150 signal. These are key experiments, and as such they really deserve to be optimized in terms of signal-to-noise ratio.

We respectfully disagree with the reviewer's characterization that the mApple/GC150 expression vector did not work in the MTZ experiments shown in Figure 6F. We see a clear increase in the GC150/mApple ratio in cells expressing NTR-BFP, indicative of death.

We agree that the image quality may not fully reflect how well GC150-P2a-mApple marks dead neurons due to the technical difficulty of this experiment. It required injection of two DNA constructs in neurons throughout the fish (pan-neuronal:NTR-BFP and pan-neuronal:GC150-P2a-mApple), and then imaging through the entire fish over multiple days to identify sporadic co-expression. As such, the expression levels of the two co-expressed constructs was difficult to control and varied substantially between different injected fish, as well as between different neurons, which is typical of transient DNA expression in zebrafish larvae.

Therefore, in an effort to improve the imaging quality, we generated a stable transgenic *neuroD:GC150-P2a-EGFP* line that provides more consistent expression of the reporter throughout larvae. We then injected transgenic larvae with a plasmid encoding *neuroD:NTR-BFP* and repeated the imaging after MTZ exposure. These images are clearer, exhibit improved signal-to-noise ratio, and are shown in the revised Figure 6F. These data confirm our conclusion that GC150-P2a-mApple marks dead neurons within live zebrafish.

Fig. S6. Showing only merged images does not facilitate understanding. Regarding the A/24h panel, it is unclear why the mCherry signal has almost disappeared. It seems that the expression vector does not really work in A panels (MTZ) compared to B panels (Tricaine), including at time 0h, which undermines observations.

In panel A/24h, the mCherry signal drop suggests that a significant number of cells are actually already dead. Measuring caspase activity at 24 hours may therefore be too late. A longitudinal kinetic study should help clarify this issue.

In the revised manuscript, this experiment appears in Fig. S9. The point of the experiment is to confirm that 24 hours after MTZ treatment, NTR-expressing motor neurons are indeed dead, despite the continued presence of mCherry in the spinal cord. We confirmed this by showing that NTR-mCherry motor neurons have a positive PhiPhiLux signal 24 hours after MTZ treatment, while DMSO-treated motor neurons do not. As suggested by the reviewer, we have now expanded Fig S9 to include the red, green and overlay channels separately, which may facilitate the observation that the PhiPhiLux signal is increased after MTZ treatment compared to DMSO

alone. We also increased the contrast of each channel, while keeping the same levels between both fish shown, so that the mCherry signal in the MTZ-exposed fish is clearer at 0 hours.

While the reviewer is correct in pointing out that the basal mCherry expression level is lower in the MTZ-exposed larvae compared to the DMSO-exposed fish, we do not believe this changes the interpretation of the experiment. Indeed, it is common for transgenic zebrafish to have differing levels of transgene expression from fish-to-fish, even within the same clutch. However, the fluorescence signal is still clearly distinguishable from the background, and the amount of NTR-mCherry in the MTZ-treated fish is clearly sufficient to mediate apoptosis in those neurons.

It is also clear that the total mCherry signal at 24 hours after MTZ treatment is reduced compared to 0 hours. The reviewer is correct that this is likely due to death of some motor neurons and clearance of fluorescent debris, which also appears to occur in Figure 5A–B and H.

Nevertheless, it is clear in the new Fig S9 that the 24 hour timepoint is not too late for detection of PhiPhiLux activity, which addresses the Reviewer's concern. As such we do not think performing a longitudinal kinetic study of PhiPhiLux activity within the zebrafish is necessary.

In this respect, it would be useful to carry out comparable kinetics regarding GC150 (Figure 6F) in order to have a idea of the time course of caspase 3 activation in connection with the cytosolic calcium peak.

Thank you for the suggestion. In response, we performed a time course comparison of the response kinetics of GC150 and a commercially available Caspase3/7 indicator. In the new Fig S3, we now show that GC150 reports death more than an hour prior to the onset of the Caspase3/7 signal.

In the same line, and in order to further validate this new cell death detection system, it is important to compare the kinetics of the GC150 signal with proven cell death markers such as TUNEL assay, which can be implement in zebrafish.

We did indeed validate the GEDI approach with a TUNEL assay in primary neurons, as shown in Fig. S2A–B. However, TUNEL has been implemented in zebrafish before. It is a terminal assay and therefore not amenable to time lapse imaging, making a time course comparison between TUNEL and GEDI signal impractical. Furthermore, although TUNEL stain is commonly used as a cell death marker in the literature, it is more appropriate to refer to TUNEL stain as a label of DNA damage, as TUNEL also will stain cells undergoing DNA repair that do not actually die (Kanoh et al., 1999). Moreover, as shown in Fig S2B not all cells with a high GEDI signal are TUNEL positive, likely because the GEDI signal precedes the onset of DNA damage in some cases. Finally, we have already demonstrated that GEDI signal marks dead neurons in zebrafish using the proven cell death marker PhiPhiLux staining (Horstick et al., 2016) in Fig S9, which directly addresses the Reviewer's concern and obviates the need for additional TUNEL staining assays.

Minor points

Fig. 3B, F: This kind of representation is somewhat difficult to understand. Histograms might be easier to handle.

Thank you for this suggestion. To address the Reviewer's suggestion and improve the visualization, we have now added parallel density plots adjacent to the original representations in Figures 3B and 3F, to illustrate the different populations of live and dead neurons as well as the shifts in populations between different neurodegenerative disease models and controls. We believe this provides an easier, more intuitive interpretation of the shifts in populations of live and dead neurons within the large datasets we report.

Fig. 5C, I: The use of red and green is confusing for the reader since they may incorrectly refer to fluorescent markers (see Fig. 5G).

Thank you for this observation. We have changed the color coding in figures 5C and I.

Reviewer #2:

Fig. 1J Separation of untreated vs +NaN3 populations is clear. The variability of GEDI ratio in the NaN3 treated population seems bigger compared to that of in vivo system (for example, Fig. 5C or I). Is this also due to the higher extracellular Ca concentration ?

There are a number of reasons the GEDI ratio can vary between experiment types. As the reviewer notes, the extracellular Ca^{2+} concentration can impact the GEDI ratio in dead neurons, as we illustrate in Supplementary Figure 1. In addition, the ratio can vary based on the particular version of GEDI, as we show in Figure 4. In Figure 5I, we used GCaMP7, which has a relatively high Ca^{2+} affinity in comparison to other GEDIs. It saturates at a much lower level of Ca^{2+} (see Figure 1A) and has the largest signal among GEDIs tested (see Figure 4D). Additional factors that can affect the magnitude and variability of GEDI signal include the imaging parameters (*i.e.*, relative brightness of the red to green channel), background present in imaging (which must be zeroed out before generating a ratio of red to green channels, and the ability to subtract the background, which can vary slightly from image to image). Additionally the NaN3 experiments were performed using an epifluorescence microscope, whereas the *in vivo* imaging was performed on a spinning-disk confocal microscope. The relative intensities of red and green channels to one another vary somewhat between these systems.

For these reasons, we are cautious about inferring differences in Ca^{2+} concentration from differences in GEDI ratio. In some of the examples above, differences in experimental design or data acquisition make it difficult to infer differences in extracellular Ca^{2+} concentration between these particular experiments based on the variabilities of GEDI ratios. On the other hand, when comparable imaging conditions are used, such as in the *in vivo* system shown in Figures 5 and 6, we can make inferences about extracellular Ca^{2+} concentration from the results. We discuss this further on the bottom of page 21:

Finally, we showed that although the RGEDI construct could not be used to detect death *in vivo*, GC150 and GCaMP7 could. Due to the differences in Ca^{2+} binding affinity between the three indicators, these data suggest that free Ca^{2+} in the extracellular spaces in brain of the zebrafish larvae is somewhat limited, and may be insufficient to induce the RGEDI fluorescence upon cell death. Thus, the difference in functionality of RGEDI in cultured neurons (Figures 1–

4) and *in vivo* in the zebrafish brain (Figures 5–6), could be due to the virtually unlimited supply of Ca^{2+} in culture medium compared to the brain, where extracellular Ca^{2+} can be limiting in times of high activity⁹⁶ and in dense synaptic areas⁹⁷. Interestingly, the limited extracellular Ca^{2+} in our *in vivo* assays raises the intriguing possibility that the sequestration of free calcium in debris of dead neurons could be a previously unexplored mechanism of cognitive decline in neurodegeneration. Future studies using 4D modeling will be key to address this question.

Related to the question above, is there any literature that Ca concentration of the extracellular fluid in zebrafish is significantly lower than 2 mM used in the culture experiments?

To our knowledge, there are no publications in which the extracellular Ca^{2+} concentration within larval zebrafish brains has been measured. This may be in part due to the small size of zebrafish larvae, which makes the use of microelectrodes to measure extracellular Ca^{2+} difficult. However, we do not expect that the overall average extracellular Ca^{2+} concentration of the zebrafish brain differs greatly from the concentrations measured in other vertebrates; in humans, the Ca^{2+} concentration is thought to be around 1.2 mM (Forsberg et al., 2019), which is lower than the 2 mM found in the neural basal media (Gibco) used in the *in vitro* experiments in this study.

There are two additional factors that may affect the extracellular Ca^{2+} concentration *in vivo* in zebrafish in comparison to *in vitro* experiments. First, isolated microdomains of Ca^{2+} occur within the vertebrate brain that vary substantially in extracellular Ca^{2+} concentrations (Verkhratsky et al., 2006), and the local accessibility of extracellular Ca^{2+} ions throughout the brain can be reduced under conditions such as tetanus activity (Nicholson et al 1978, Heinemann et al 1990, Smith 1992, Montague 1996). Thus, there is likely to be large variability in the extracellular Ca^{2+} concentration in different regions of the brain, which would be expected to affect the ability of GEDIs, with especially low Ca^{2+} affinities, to saturate and reach peak fluorescence.

Secondly, the small size of the zebrafish brain means that there are far fewer total Ca^{2+} ions in the extracellular fluid than are available *in vitro*. This is particularly important as neurons die and GEDI signal is elevated throughout the brain, such as in Figure 5. The high Ca^{2+} signal within dead neurons indicates that extracellular Ca^{2+} has been sequestered and is no longer an available part of the extracellular pool, further driving down the total Ca^{2+} . In contrast, in *in vitro* experiments with superfluous extracellular media, the effect of sequestered extracellular Ca^{2+} within neurons would be negligible. We believe the combination of the presence of microdomains with varying extracellular Ca^{2+} concentrations, the reduced amount of total Ca^{2+} in the extracellular pool, and the sequestration of extracellular Ca^{2+} in dead neurons contributes to the inability of RGED1-P2a-EGFP to function properly in our *in vivo* studies compared to GC150-P2a-EGFP, which has a high Ca^{2+} binding affinity and was used successfully.

#p10 L2 Fig 2C should be Fig2D.

Thank you for carefully noting the mistake. We have made the correction.

#Fig2E How does the decrease of fluorescence compare to the disintegration of the plasma membrane, with regards to the time course? In other words, how much of the decrease is caused by the active degradation as opposed to the simple diffusion into the extracellular space?

In tissue culture, we often see fluorescence remain in place for up to 168 hours after death (Figure 2E), suggesting the fluorescent protein persists within the plasma membrane of the cell and does not diffuse into the surrounding media. However, there is heterogeneity in this process, as we sometimes observe fluorescence disappear fairly rapidly after neuronal death as indicated by GEDI signal. Both *in vivo* in zebrafish and in primary neuronal culture, glial cells capable of actively degrading fluorescent proteins are present, and could be involved in accelerating degradation of the fluorescence signal by disintegrating the plasma membrane to the point at which all of the fluorescence can quickly diffuse into the extracellular media. Amongst those cells in which significant fluorescence remains, we also see a gradual degradation of fluorescence signal from both EGFP and RGEDI after death at a relatively steady rate (Figure 2E). We suspect this degradation of fluorescence signal is the result of the combination of active proteases within the cell debris that gradually break down the EGFP and RGEDI proteins, leakage and diffusion of some of that protein into the extracellular space, and/or gradual bleaching of fluorescent proteins.

#Fig. 3G Neurons showed lower survival rate in the ~200 hr culture period. Given the time course of the disease in human patients, the rapid change observed in the experiment is somewhat unexpected. Unlike the experiments in Fig. 3A-C, genetic factors causing the disease is expected to be similar to those in patients from whom iPSCs are derived. Do authors have any idea to explain this difference?

Thank you for the comment. To clarify, there were errors on the y axis of both Figures 3C and 3G which have now been corrected, though the overall results and trends within the data have not changed.

Variability in the absolute magnitude of our cumulative risk-of-death measurements in murine and iPSC cultures is something we have frequently encountered in our experience using this statistic with more conventional measurements of death for more than 15 years (Arrasate et al., 2004, Barmada et al., 2010, Skibinski et al., 2014). There are many differences between the neurons imaged in Figures 3C and 3G, including different disease models, species (rat vs. human), neuron types (primary cortical neurons vs. human iPSC-derived neurons), heterogeneity of the culture (diverse cell and neuron types from rat cortex vs. more homogenous motor neuron differentiations), differences in media composition (which based on match to the cell type may give more or less viability), and genetics of disease-associated proteins (transfection vs. mutation in the genome). In our past experiences with cumulative risk-of-death measurements we have found each of the factors can affect the viability of the culture. We have also found that with careful measures to control for factors that can affect variability such as using appropriate in-plate, and otherwise matched controls to which we make direct comparisons, we can deduce the effects that can be attributed to the genetic or pharmacological perturbation we are studying. In the case of the neurons in Figure 3A–C and 3G, because differences were not controlled for between experiments, making direct comparisons across experiments is difficult to do. Nevertheless, a comparison of the control condition from Figure 3C (RGEDI-P2a-EGFP) to the control line in Figure 3G shows that 168 hours post transfection, a higher proportion of rat cortical neurons than iPSC-derived motor neurons appears to still be alive, suggesting that differences between the two experiments are not due solely to the disease modeling conditions. One clear factor that could be contributing to the overall reduced survival rate of human iPSC-

derived neurons could include the lack of support cells such as pre-myelinating Schwann cells, which have been shown to increase survival of cultured iPSC-derived motor neurons (Suh et al., 2017).

#Fig.4D In the second paragraph of p13, authors state that “As expected cells expressing GC150-P2a-mApple showed increased GEDI ration.” Is GC150-P2a-mApple compared to RGED1-P2a-3XBFP here? I am confused because the GECI and the reference are both changed. Please clarify.

Thank you for alerting us to this confusion. To clarify our point that the increasing GC150-P2a-mApple signal indicates death, we added a new control to Figure 4D in which neurons are expressing EGFP-P2a-mApple. As expected, we find that no increase in EGFP/mApple signal is detected in neurons expressing EGFP-P2a-mApple after exposure to NaAz, and this provides us a baseline from which statistical comparisons of the red:green ratio can be made in order to demonstrate that each version of GEDI does indeed report death.

P15 l3 “mnx:Gal4; UAS:mCherry; UAS:EGFP zebrafish larvae “ seems to be a mistake. It should contain NTR under UAS.

#p16 l3 GcaMP7 needs to be in capital letters.

We thank the reviewer for a careful reading of the manuscript and finding these mistakes. We have corrected the text with “UAS:NTR-mCherry” and “GCaMP7” in each case.

#Fig.5B-D The immobility of 24 hr larvae in the absence of Ca increase is highly unexpected and important. Authors discuss the implication of this finding on p20. Please expand this discussion so as to include possible mechanisms leading to the axon retraction in the absence of Ca increase.

We do in fact see an increase in Ca^{2+} in motor neurons of immobile larvae 24 hours after MTZ treatment, as we show in figure 5H–I. At the same time, we do not detect axon retraction at that timepoint (Figure 5C). At 48 hours post MTZ, we see both axon retraction and Ca^{2+} increase. We do not suggest that death can occur in the absence of Ca^{2+} increase, but rather suggest that a Ca^{2+} increase to a certain threshold that is detected by GEDIs can be used as a readout of neuronal death. Furthermore, we do not intend to suggest that axon retraction is a Ca^{2+} -independent phenomenon.

On p20 we have clarified by explaining:

Finally, we showed that although the RGED1 construct could not be used to detect death *in vivo*, GC150 and GCaMP7 could.

Thus, although we do not see a Ca^{2+} increase to the level that RGED1 is activated, we do see an increase in Ca^{2+} levels in immobile larvae, 24 hours after incubation in MTZ.

Reviewer #3:

1) On p7, the authors indicate that the increase in GEDI ratio preceded obvious morphology

changes. However, Fig. 1H does not show that clearly. Either a better image is needed here or the authors should clarify this statement.

We thank the reviewer for this insightful comment. In response we have created a separate high contrast EGFP image for each time point in Figure 1H that more clearly shows the morphology of neuron #1, in which the GEDI ratio increase precedes obvious morphology changes. We have also added a yellow asterisk to further indicate the timepoint at which neuron #1 appears to have normal morphology and visible neurites even though its GEDI ratio is already elevated.

2) Figure 5 showing the capability of using GCAMP7 as a death indicator – but only under conditions when the zebrafish are immobilized seems a bit out of place given the focus of the rest of the manuscript on these new GEDIs. It seems highly unlikely that GCAMP7 will be used as a death indicator because of the need to block neuronal activity. There is value in making this point and perhaps the authors could consider moving this figure to the supplement.

Based on this suggestion, we have moved Figure 5 to become Supplementary Figure 9. We appreciate the reviewer's suggestion that GCaMP7 is less useful as a death indicator than other GEDIs- disambiguating neuronal activity from the high calcium levels associated with death was the main impetus behind using low affinity Ca^{2+} indicators as GEDIs. However, paralytic conditions such as incubation in tricaine are still commonly used for imaging live zebrafish despite their inhibition of neuronal activity. So, measuring cell death using transgenic GCaMP lines, which are commonly available throughout the zebrafish community, may not be as unlikely as the reviewer suggests.

Furthermore, we show in the newly added Supplementary Figure 5 that GEDIs can detect cell death in other non-neuronal cell types such as HEK293 cells. Therefore, we expect cell death detection with GCaMP7 could be especially useful and sensitive in other cell types that do not typically show Ca^{2+} fluctuations, and could be detected in zebrafish under other paralytics that do not anaesthetize the animal.

Finally, beyond the novelty of demonstrating the efficacy of the commonly used GCaMP as a cell death indicator, the data presented in Supplementary Figure 9 were also important for advancing our understanding of how GEDIs work *in vivo*. In the current Figure 5, we show that that the lowest Ca^{2+} affinity GEDIs such as RGEDI may not work *in vivo* because extracellular Ca^{2+} levels may be limited. Because of this, it was critical for us to establish that, in principle, Ca^{2+} dyshomeostasis could be detected *in vivo* as well as *in vitro*. These experiments using GCaMP7 both establish that Ca^{2+} dyshomeostasis can be detected *in vivo*, and that higher affinity Ca^{2+} indicators such as GCaMP7 and GC150 in Figure 5 will be preferable in that context.

3) The authors state in the abstract that the GEDIs detect Ca levels that cells achieve early in their death process. However, as shown by the authors, the GEDIs detect Ca that enters the cell after the plasma membrane integrity is lost. Therefore, these indicators are essentially detecting loss of plasma membrane integrity, which would not be considered an early event in the cell death process. The language here would need to be corrected.

We appreciate the reviewer's insight. In our initial draft, we used imprecise language that suggested GEDIs detect the loss of plasma membrane integrity. However, we now believe that

message is contrary to the overwhelming evidence on how GEDIs detect neuronal death. Although we show in Supplementary Figure 1 that influx of extracellular Ca^{2+} is necessary to drive GEDI signal, the fluorescent proteins remain largely in place after GEDI signal is activated rather than diffusing out into the extracellular media, indicating that the plasma membrane is intact.

Furthermore, rather than detecting a late stage of membrane integrity breaking down, we now show in Supplementary Figure 2 as well as the newly added Supplementary Figure 3 that GEDI indicates death earlier than any other indicator we have tested. GEDI signal appears before other indicators of death such as Caspase3/7, Propidium Iodide, and Ethidium Homodimer, which we believe indicates that GEDI does indeed mark cells early in the death process.

To be more specific in our language, we have now changed the language in the manuscript to more specifically indicate that GEDI detects an irreversible loss of Ca^{2+} homeostasis. Thus, the sentence referencing plasma membrane integrity on page 9 has been modified to the following:

In principle, GEDI should be able to detect all cell death events, as it detects loss of Ca^{2+} homeostasis rather than a specific substrate of a cell death pathway.

Reviewer #4:

Although the authors claimed development and application of a GEDI, reality of the “GEDI” is an already reported GEI “CEPIA” itself and versatility of the detection for different cell types is questioned. Overall, this manuscript holds difficulty for publication from this journal, because it does not adequately represent the perceived importance of the technology.

Major points:

1. The authors developed GEDI from RCEPIA by removing the ER retention signals. However, the following explanation in the original paper, “We expressed the colour variants of CEPIA with ER-targeting signal sequences in HeLa cells”, indicates that “CEPIA” stands for the variants without target signal sequence. To avoid confusion, different names CEPIA and GEDI should not be used for the same indicator. The term “GEDI” should be used to refer to a broad range of genetically encoded death indicator including GCaMP6-150, but not to refer to RCEPIA. The title of the manuscript must be also toned down to reflect this work is one of the applications of the low affinity GEI.

We respectfully disagree with the reviewer’s sentiments that GEDI is not the appropriate name for the biosensor that we have developed and characterized in this manuscript, and that we have inadequately represented the importance of this technology. As the reviewer notes and as we mention on the bottom of page 6, we did indeed develop GEDI from RCEPIAer. However, we strongly dispute the reviewer’s characterization of the original report of CEPIA (Suzuki et al.

2014), particularly that “CEPIA” stands for variants without target signal sequence. In fact, that report notes the abbreviation CEPIA stands for, “calcium-measuring organelle-entrapped protein indicators”. A CEPIA without an ER-targeting signal (RGEDI) would no longer be an organelle-entrapped protein. Therefore, it would be inappropriate for us to call a sensor CEPIA if it is no longer targeted to an organelle because its ER-targeting signal has been removed.

We also believe the quote cited from Suzuki et al., is misinterpreted by the reviewer. The reviewer suggests that because the Suzuki et al., report cites “color variants of CEPIA with ER-targeting signal sequences”, it means that “CEPIA” refers to a base protein to which organelle targeting sequences can be added. However, CEPIA does not refer to one protein. Some CEPIAs described in Suzuki et al. contain mitochondria(mt)-targeting sequences and are engineered to measure Ca^{2+} fluctuations found in mitochondria, while others contain ER-targeting sequences and are engineered to detect Ca^{2+} fluctuations found in the ER. Importantly, the CEPIAer and CEPIAmt proteins were independently developed by site-directed mutagenesis on previously developed cfGCaMP2/GECO calcium indicator proteins targeted to the ER and mt respectively. Thus, the CEPIA in CEPIAer and the CEPIA in CEPIAmt have different protein sequences outside of their organelle targeting, making them actually independent daughter proteins derived from a GCaMP/GECO hybrid protein. Within this context, the quote “color variants of CEPIA with ER-targeting signal sequences” is actually differentiating the type of CEPIA (engineered for mt or ER Ca^{2+} levels) rather than a base CEPIA protein to which an ER-targeting signal is affixed. Therefore, CEPIA should not be used to refer to a common protein without the targeting sequence, and it would be misleading for us to describe GEDI as CEPIA because it would not indicate which CEPIA variant (er or mt) was being used.

While we appreciate the suggestion of using the name “low-affinity GECI”, we feel that that name is less specific for the application we describe in the manuscript. Traditionally, biosensors are named for their function rather than their relative biophysical properties, and just as the abbreviation CEPIA denotes the sensor’s function of measuring Ca^{2+} levels of organelles, we believe the name of the death indicators we generated should indicate their utility in detecting cell death.

Although we understand the sentiment that CEPIA proteins and GEDI proteins should be differentiated to avoid confusion, we have made an effort throughout the manuscript to be consistent in the naming scheme of all of the GEDIs we characterize. The only instance in the manuscript where both names refer to the same indicator is in Figure 1A, where we show the comparative relative fluorescence of Ca^{2+} binding. Since both proteins have the same kinetics, it is difficult to distinguish them in this circumstance. But to avoid confusion, we removed the “/” separating RGEDI and RCEPIA1er, so that there is no confusion as to whether they are the same protein.

Finally, we addressed the reviewer’s suspicion of the versatility of this detection method for other cell types other than neurons. All cell types are thought to maintain a homeostatic gradient with extracellular Ca^{2+} , and thus detection of the loss of Ca^{2+} homeostasis by GEDIs should be adaptable across different cell types. To demonstrate this concept, we have now added a Supplementary Figure 5, which shows the efficacy of detecting cell death in a non-neuronal, non-excitable cancer cell line, HEK293. Thus, we believe our manuscript adequately represents the perceived importance of the technology in the ability of GEDIs to acutely detect cell death in live imaging experiments.

2. Intensity of EGFP and RGED1 seems saturated at the high intensity region in the Figure 1H. If the saturated values are used for the calculation of the graph in Figure 1I, the graph contains significant error. In addition, intensity ratio image instead of the overlay image is preferable to compare with the graph in Figure 1I.

Thank you for raising this issue. We were careful prior to imaging to set imaging parameters such that we did not achieve fluorescence levels that would saturate pixels on the camera. Our cameras have a 16 bit dynamic range which helps ensure we keep pixel values in a range so that they are not susceptible to floor or ceiling effects. But that range is also larger than we can display on computer screens or in printed images, which are typically limited to 8 bit. This means that the contrast in the image display must be changed in order to see some dim features of the images such as neurites, which have much dimmer fluorescence than the neuronal soma. When the contrast is enhanced on the display, it has the effect of making the brighter regions of the image such as the soma appear to saturate the pixels in the display. Nevertheless, measurements were obtained from images in which saturated values were not present.

In this case, we thought it was necessary to display both the dim neurites and the bright soma from the neurons in Figure 1H. Neurites are often used as a defining morphological feature of live neurons as they are typically degraded when a neuron dies, and it is necessary to enhance the contrast in the images in order to display those neurites, as we did for Figure 1H. The reviewer is correct that enhancing the contrast in the micrograph has the added effect of making EGFP expression within the soma look saturated. However, the raw image signal, from which we generate the quantification shown in Figure 1I, did not contain any artifacts associated with saturated signal, and we have now added a low contrast panel to Figure 1H to demonstrate that pixels were not saturated.

3. Since the contents of the section of the “GCaMP acutely reports death in vivo” is out of the concept of this manuscript, “dead-cell detection by low affinity GEC1”, it should be removed from the main text and described in the supplementary note if the authors want. Moreover, the indicator for the zebrafish specific artificial condition is not so much attractive for a broad range of readers who are interested in cell-death imaging.

Based on these recommendations, we have moved what was formerly Figure 5 to become Supplementary Figure 9, and we have removed the section heading “GCaMP acutely reports death *in vivo*”. Nevertheless, we disagree that the section in which we describe using GCaMP as a reporter for cell death in zebrafish is inconsistent with the main message of this manuscript. We demonstrate in Figures 1, 4, and Supplementary Figure 9 that GCaMP can detect neuronal death, a fact that has not been previously demonstrated in the literature to our knowledge. We believe this is an important point to make given that the GCaMP family of indicators is commonly used throughout the field of neuroscience, and because GCaMP detects death by the same mechanism as other GEDIs—detection of the loss of Ca^{2+} homeostasis within the neuron—it fits well within our story.

Moreover, it is very common practice to image live zebrafish under tricaine anesthetic immobilization. While the audience that engages in zebrafish live imaging may not be as large as that interested in cell-death imaging in general, we still believe it is important to demonstrate

that a biosensor as commonly used in zebrafish as GCaMP can be used as a GEDI when other GEDIs are not available.

Beyond the novelty of demonstrating the efficacy of GCaMP as a cell death indicator, our descriptions of GCaMP detecting cell death fit well within the concept of the manuscript for several other important reasons. First, the use of GCaMP in Supplementary Figure 9 allowed us to establish in Figure 5 that the lowest Ca^{2+} affinity GEDIs such as RGEDI may not work *in vivo* if extracellular Ca^{2+} levels are limited. Because of these circumstances, it was critical to establish whether GCaMP signal was elevated in dead neurons *in vivo* to determine if loss of Ca^{2+} homeostasis occurs *in vivo* as well as *in vitro*.

Second, we have new data indicating GCaMP could act as a better GEDI indicator in other cell types besides neurons, which we have added. We do note in Figure 1 that GCaMP signal indicating death can be confused with typical calcium transients that GCaMP reports in neurons. However, in the many cell types that do not generate large Ca^{2+} transients, GCaMP could potentially be used as a more sensitive death detection tool. Related to this point, we have added a new Supplementary Figure 4 in which we demonstrate the effectiveness of death detection by RGEDI-P2a-EGFP in HEK293 cells. Characterizing and optimizing GEDI indicators in non-neuronal cells is beyond the scope of this manuscript. However, in cell types such as HEK293 cells in which activity-dependent Ca^{2+} transients do not occur, it is entirely possible that GCaMP could be a more effective death indicator.

4. Different sensitivity of GFP and RFP to lysosomal proteases is mentioned as a key factor for developing tandem tag biosensors to measure autophagy and Reference-46 is cited for supporting that (p. 10 L8). However, according to this paper, the sensor was developed based on the “different pH stability of EGFP and mRFP”, and the word “protease” could not be found by my word search for this paper.

We thank the reviewer for noting the incorrect citation and terminology used in the manuscript. We have now corrected the manuscript with the appropriate reference (Kimura et al., *Autophagy*, 2007), which includes the following reference to the differential regulation of the stability of EGFP and mRFP by lysosomal hydrolases. We have also changed “lysosomal proteases” to “lysosomal hydrolases”.

“Here we found that a fusion protein of monomeric red-fluorescence protein and LC3, the most widely used marker for autophagosomes, exhibits a quite different localization pattern from that of GFP-LC3. GFP-LC3 loses fluorescence due to lysosomal acidic and degradative conditions but mRFP-LC3 does not, indicating that the latter can label the autophagic compartments both before and after fusion with lysosomes... This result indicates that the fluorescence from luminal GFP-

LC3 was attenuated in the lysosomal acidic environment and that the protein was degraded by lysosomal hydrolases."

5. Although the relative dimness of BFP is mentioned as a reason of the smaller dynamic range of RGEDIP2a-3xBFP (p. 13 L4), the intensity difference of the reference fluorescence does not cause a difference in the dynamic range in principle. A more satisfactory explanation is required.

Thank you for alerting us to this error. We have removed the reference to the smaller dynamic range of RGEDIP2a-3xBFP as being due to the relative dimness of BFP. The reviewer is correct in pointing out the dim 3xBFP cannot fully explain why the RGEDIP2a-3xBFP dynamic range is smaller than in RGEDIP2a-EGFP. An alternative explanation for the reduced dynamic range is that the RGEDIP2a-3xBFP protein is not as efficiently translated as the RGEDIP2a-EGFP protein, resulting in lower expression 3xBFP within the neuron and fluorescence that is so dim that it approaches the noise floor. Typically, if the RGEDIP2a-EGFP protein has reduced expression within a cell, there will be less of an increase in red fluorescence signal from RGEDIP2a after binding high levels of Ca^{2+} in dead cells, but that smaller signal will get normalized by reduced reference fluorescence in the denominator of the GEDI ratio (RGEDIP2a/EGFP). However, if the 3xBFP is so dim that its signal is below the noise floor and noise artificially inflates the normalization factor in the GEDI ratio, then it may appear to reduce the dynamic range of RGEDIP2a-3xBFP. Although we do show signal from RGEDIP2a-3xBFP has a reduced dynamic range compared to RGEDIP2a-EGFP in Figure 4D, we have also added an additional control of EGFP-P2a-mApple that indicates that RGEDIP2a-3xBFP signal does significantly increase in comparison to non-GEDI signal.

6. Response of the GC150-P2a-mApple to the field stimulation should also be checked to confirm whether its signal change is truly cell-death specific.

To address this question, we have added Supplementary Figure 7 to the manuscript. In it we demonstrate that neurons transfected with GC150-P2a-mApple are not responsive to field stimulation of 30Hz compared to neurons transfected with GCaMP6f-P2a-mRuby. At the same time, we demonstrate that both GC150-P2a-mApple and GCaMP6f-P2a-mRuby detect the onset of death after application of a toxin.

Minor points:

1. The frequency of the field stimulation in Figure 1B was described as 10 Hz (p. 6 L1 from the bottom) in the main text. However, it was 30 Hz in the Method section (p. 23 L7 from the bottom) and the figure legend (p. 27 L1 from the bottom). This discrepancy should be corrected.

We thank the reviewer for the careful reading of the text and catching this mistake. We have corrected the text to read 30 Hz throughout the manuscript.

2. Since NaN3 has already been used in the description for the earlier experiment, an explanation about NaN3, “a known cytotoxin”, (p. 8 L4) might be unnecessary.

We thank the reviewer for catching this redundancy. We have moved the description of NaN3 as a known cytotoxin up to the first instance and description of NaN3 in the text.

3. At p.11 L5-6 from the bottom, the authors mentioned “a clear separation of live and dead neurons could still be observed” for Figure 3B. However, there is no other way than the reference line drawn by manual curation to discriminate live and dead neurons in this figure. Clear discrimination based on the data is required. This is also true of Figure 3F.

To clarify the separation between populations of live and dead neurons, we added density plots to Figures 3B and 3F. These additional data descriptions obviate the separation of live and dead populations of neurons as well as display differences in the total quantities of live and dead neurons per experiment across all timepoints.

4. The authors compared the rate of signal increase after exposure to NaN3 between RGEDI-P2a-3'BFP and RGEDI-P2a-EGFP or GCaMP6f-P2a-mRuby58. However, it is difficult to compare the rates from Figure 4D. Independent graph, which can directly show statistical significance, is required.

Thank you for the insightful comment. We have added a new subpanel to Figure 4D comparing the rates of signal increase across GEDIs as well as an ANOVA statistical analysis that shows no statistical difference in the rate of increase of each GEDI.

5. The graph to show kinetics of the responses of GEDI (τ value), which can directly show statistical significance, must support description at p.13 L2-4 from the bottom.

In adding a graphical illustration of the τ rates of signal increases and their statistical analysis, we further show support for the description at p.13, L2–4 from the bottom.

6. “Figure 5D” at p. 15 L6 might be “Figure 5D-F”.

Thank you for noting this omission. We have corrected the callout to Figure 5D–F.

7. At p. 20 L9, loss of what fluorescence does the authors mean?

Thank you for the noting this confusing statement. We have amended the sentence to state, “loss of fluorescence from neurons expressing mCherry”.

8. At p. 23 L8 from the bottom, what the value 0.95 stand for? Numerical aperture?

Thank you for noting the omission. We have indicated 0.95 numerical aperture in the text.

9. In Figure 1J, although there are blue plots in “0 Minutes”, no explanation for that in the figure legend. Green color might be correct.

Thank you for this useful suggestion. We have added the following text to the caption of Figure 1J to clarify that the blue dots in 0 minutes represent a different well than the green and red dots in the same figure:

Quantification of GEDI ratio in rat primary cortical neurons before (blue dots) and from a separate well 5 minutes after NaN_3 -induced neuronal death (green dots = live, red dots = dead).

REVIEWER COMMENTS

Reviewer #1 (Remarks to the Author):

on the whole, the authors have responded satisfactorily to the remarks

Reviewer #2 (Remarks to the Author):

Authors addressed all of my concerns sufficiently.

Reviewer #4 (Remarks to the Author):

The minor points in the previous review comments were properly revised in the current manuscript. However, there are still some disagreements with the reviewer on the main subject of the paper. Further revision is required to address that.

1. Since the intention of the sentence in the previous reviewer comment, "The term "GEDI" should be used to refer to a broad range of genetically encoded death indicator", was not well noticed by authors, I explain further detail. GEI is an abbreviation of the Genetically Encoded Ca²⁺ Indicator that does not stand for a specific indicator instead that holds any types of indicators including CEPIA, GCaMP, GECO, and Cameleon, etc. Similarly, GEVI is an abbreviation of the Genetically Encoded Voltage Indicator that stands for a broad type of indicator. GEDI should be used as a generic name for genetically encoded death indicators in the same vein rather than to indicate an individual indicator.

2. To make clear the difference of contrast of images in Figure 1H. look-up table to represent the relationship between intensity value and image color brightness should be shown.

3. If the term "GEDI" is for the generic name of genetically encoded death indicator, GCaMP with high affinity used in Supplementary Figure 9 can be accepted as GEDI in the broad definition. Meanwhile, if the GEDI stands for the specified indicator derived from the CEPIA, the concept of the paper is limited to that and usage of GCaMP must be out of the scope. Therefore, although there is some importance, the detailed description related to the GCaMPs especially results obtained by GCaMP7 should be moved to Supplementary Notes to support CEPIA based indicators.

4. As a citation of Kimuraz's paper in the response comment from the authors, "GFP-LC3 loses fluorescence due to lysosomal acidic and degradative conditions but mRFP-LC3 does not", the difference of pH stability between EGFP and mRFP is also a key factor for the development of tandem tag biosensors. That should be explained.

5. Data in Fig. 3B, C, F and, G were changed from the previous manuscript without any notification to the reviewers. Only the correction of errors on the y-axes of Fig. 3C and G was described in the response comment to Reviewer #2. There are unclear things in the data of new graphs. In Fig. 3C data point at 160 hours is missed for Htt Ex1 Q75 and Htt Ex1 Q25. The distribution of the points in Fig. 3F seems significantly different.

Authors' Response to Reviewers' Comments

We thank the reviewers for their careful reading of our manuscript. Please find below a point-by-point response to the concerns raised during the initial review. The manuscript has now been thoroughly revised accordingly, and we hope that it is now acceptable for publication.

Reviewer #4 (Remarks to the Author):

The minor points in the previous review comments were properly revised in the current manuscript. However, there are still some disagreements with the reviewer on the main subject of the paper. Further revision is required to address that.

1. Since the intention of the sentence in the previous reviewer comment, “The term “GEDI” should be used to refer to a broad range of genetically encoded death indicator”, was not well noticed by authors, I explain further detail. GECl is an abbreviation of the Genetically Encoded Ca²⁺ Indicator that does not stand for a specific indicator instead that holds any types of indicators including CEPIA, GCaMP, GECO, and Cameleon, etc. Similarly, GEVI is an abbreviation of the Genetically Encoded Voltage Indicator that stands for a broad type of indicator. GEDI should be used as a generic name for genetically encoded death indicators in the same vein rather than to indicate an individual indicator.

Thank you for the insightful comment. We fully agree that GEDI should be used as a generic term for any of these indicators, in the same vein as FlaSh is part of a generic family of genetically encoded voltage indicators (GEVI). The title of the manuscript is “Genetically encoded cell-death indicators (GEDI) to detect an early irreversible commitment to neurodegeneration”, rather than referencing a single GEDI indicator. Throughout the manuscript, we do indeed use the term GEDI to refer to the broad range of genetically encoded death indicators including RGEDI-P2a-EGFP, RGEDI_{nls}-P2a-EGFP_{nls}, RGEDI-P2a-BFP, GC150-P2a-mApple, and GC150_{nls}-P2a-mApple_{nls}. Furthermore, the following paragraph from the last paragraph of the introduction makes clear that we consider the term GEDI to indicate a general term rather than a specific indicator:

Here, we introduce a new class of GECIs for the detection of cell death in neurons that we call Genetically Encoded Death Indicators (GEDIs). We show that GEDIs can robustly indicate the moment when a neuron's ability to maintain Ca²⁺ homeostasis is lost and cannot be restored, providing an earlier and more acute demarcation of the moment of death in a degenerating neuron than previously possible.

Nevertheless, due to the difficulty in pluralizing the GEDI abbreviation, and the narrative rollout from an initial GEDI indicator (RGEDI-P2a-EGFP) to the later introduction of a family of

similar indicators, we can see that in some cases we used the singular GEDI rather than the plural GEDIs, which may have led to some confusion. To clarify any confusion around the use of GEDI as a general term for a class of indicators, we have added an “s” and pluralized the term GEDI throughout the manuscript. For instance we have changed the abstract to include the following pluralized use of the term GEDIs (bold and underlined characters have been added):

*Here, we report the development of a new family of fluorescent biosensors called **genetically encoded death indicators** (GEDIs). GEDIs specifically detect an intracellular Ca^{2+} level that cells achieve early in the cell death process and marks a stage at which cells are irreversibly committed to die.*

2. To make clear the difference of contrast of images in Figure 1H, look-up table to represent the relationship between intensity value and image color brightness should be shown.

We have added lookup calibration bars to Figure 1H.

3. If the term “GEDI” is for the generic name of genetically encoded death indicator, GCaMP with high affinity used in Supplementary Figure 9 can be accepted as GEDI in the broad definition. Meanwhile, if the GEDI stands for the specified indicator derived from the CEPIA, the concept of the paper is limited to that and usage of GCaMP must be out of the scope. Therefore, although there is some importance, the detailed description related to the GCaMPs especially results obtained by GCaMP7 should be moved to Supplementary Notes to support CEPIA based indicators.

What may be causing some confusion is the use of the term RGEDI within the manuscript to refer to the first GEDI we generated. The individual indicator RGEDI stands for the Red Genetically Encoded Death Indicator. To generate RGEDI we reengineered an ER-localized RCEPIAer indicator to become RGEDI. Previously, the reviewer seemed to take issue with our renaming of the new protein RGEDI, arguing that “[the] reality of the “GEDI” is an already reported GECI “CEPIA” itself and versatility of the detection for different cell types is questioned.” As we pointed out, RGEDI is a novel and unique reporter and is not the same as RCEPIA. Furthermore, other than GEDIs that contain RGEDI and RGEDInls, none of the other GEDIs we describe in the manuscript have anything to do with CEPIAs. During our previous response, we adapted the reviewer’s above language in referring to RGEDI as GEDI in responding to the difference between RGEDI and RCEPIA. That may have inspired further confusion about the term GEDI referring to a particular indicator rather than a general class of indicators. Nevertheless, throughout the manuscript we consistently make use of GEDI as a general class of indicator and RGEDI as a specific indicator within that class.

We fully believe that the high affinity GCaMP (referred to as GC150 within the manuscript) is a part of the general GEDI family of indicators. In reality, GC150 and GCaMP7 have absolutely

no relation to CEPIAs as they were independently derived. Therefore it would also be inappropriate for us to only mention GCaMP7 and GC150 in support of CEPIA based indicators.

4. As a citation of Kimuraz's paper in the response comment from the authors, "GFP-LC3 loses fluorescence due to lysosomal acidic and degradative conditions but mRFP-LC3 does not", the difference of pH stability between EGFP and mRFP is also a key factor for the development of tandem tag biosensors. That should be explained.

We have revised the sentence citing Kimuraz's paper in the manuscript to read the following:

It is known that GFP and RFP are differentially sensitive to lysosomal hydrolases due to the difference in pH stability of GFP and RFP, and that differential sensitivity has been exploited to develop tandem tag biosensors to measure autophagy^{46,47}

5. Data in Fig. 3B, C, F and, G were changed from the previous manuscript without any notification to the reviewers. Only the correction of errors on the y-axes of Fig. 3C and G was described in the response comment to Reviewer #2. There are unclear things in the data of new graphs. In Fig. 3C data point at 160 hours is missed for Htt Ex1 Q75 and Htt Ex1 Q25. The distribution of the points in Fig. 3F seems significantly different.

We apologize for the confusion in Figure 3 around changes that were not mentioned in the response to reviewers. In the process of remaking plots in Figure 3B, C, F, and G to include companion density plots, we noticed slight errors in the plots that were corrected in the revision. One thing we noticed was that a data point from 160 hours for Htt Ex1 Q97 contained no data. When we tracked back to the images, we found that for that condition and timepoint, the images were blank. This tends to occur when the robotic microscope is unable to align images to the previous timepoint as there are no cells left to align. But while we previously reported that time point as having no increase in death, we felt it was more correct to censor that timepoint from both the Htt Ex1 Q97 and Htt Ex1 Q25 timepoint as we could no longer see any cells for the control condition. We modified the plot from Htt Ex1 Q97 and Htt Ex1 Q25 to end at 144 hours.

In Figure 3F, we simply flipped the color scheme between control and SOD1 D90A to match G and the density plot in 3F. When generating the density plot, we also found that the scale on the previous plot had been artificially cropped, eliminating a small minority of very high GEDI

ratio values. We have since expanded the y axis to include those previously excluded datapoints.

We apologize for the previous errors in our plots as well as the confusion around our corrections to the plots. Nevertheless, all trends in the data are the same in Figure 3 between each draft.